# Comparison and evaluation of anthropogenic emissions of $SO_2$ and $NO_x$ over China

Meng Li[1,3,a], Zbigniew Klimont[2], Qiang Zhang[1], Randall V. Martin[4], Bo Zheng[3], Chris Heyes[2], Janusz Cofala[2], Yuxuan Zhang[1,a] and Kebin He[3]

[1]Ministry of Education Key Laboratory for Earth System Modeling, Department for Earth System Science, Tsinghua University, Beijing, China

[2]International Institute for Applied Systems Analysis (IIASA), Laxenburg, 2361, Austria

[3]State Key Joint Laboratory of Environment Simulation and Pollution Control, School of Environment, Tsinghua University, Beijing, China

[4]Department of Physics and Atmospheric Science, Dalhousie University, Halifax, Canada

[a] now at Max-Planck Institute for Chemistry, Mainz, Germany

*Correspondence to*: Meng Li (M.Li@mpic.de) and Bo Zheng (bo.zheng@lsce.ipsl.fr )

**Abstract.** Bottom-up emission inventories provide primary understanding of sources of air pollution and essential input of chemical transport models. Focusing on $SO_2$ and $NO_x$, we conducted a comprehensive evaluation of two widely-used anthropogenic emission inventories over China, ECLIPSE and MIX, to explore the potential sources of uncertainties and find clues to improve emission inventories. We first compared the activity rates and emission factors used in two inventories, and investigated the reasons of differences and the impacts on emission estimates. We found that $SO_2$ emission estimates are consistent between two inventories (with 1% differences), while $NO_x$ emissions in ECLIPSE's estimates are 16% lower than those of MIX. The FGD (Flue-gas Desulfurization Devices) penetration rate and removal efficiency, LNB (low-$NO_x$ burners) application rate and abatement efficiency in power plants, emission factors of industrial boilers and various vehicle types, and vehicle fleet need further verification. Diesel consumptions are quite uncertain in current inventories. Discrepancies at sectorial and provincial level are much higher than those of the national total. We then examined the impacts of different inventories on model performance, by using the nested GEOS-Chem model. We finally derived top-down emissions by using the retrieved columns from the Ozone Monitoring Instrument (OMI) and compared with the bottom-up estimates. High correlations were observed for $SO_2$ between model results and OMI columns. For $NO_x$, negative biases in bottom-up gridded emission inventories (-21% for MIX, -39% for ECLIPSE) were found compared to the satellite-based emissions. The trends from 2005 to 2010 estimated by two inventories were both consistent with satellite. The inventories appear to be fit for evaluation of the policies at an aggregated or national level, more work is needed in specific areas in order to improve the accuracy and robustness of outcomes at a finer spatial and also technology level. To our knowledge, this is the first work where source-sector comparisons detailed to technology-level parameters are made along with the remote sensing retrievals

and chemical transport modeling. Through the comparison between bottom-up emission inventories and evaluation with top-down information, we identified potential directions for further improvement in inventory development.

## 1    Introduction

$SO_2$ and $NO_x$ are important precursors of secondary $PM_{2.5}$, contributing to severe environmental problems including haze and acid rain, and have been shown detrimental to human health and ecosystems (Seinfeld and Pandis, 2006). China's anthropogenic emissions have become one of the major contributors to the global budget during the last decade (Klimont et al., 2013; Hoesly et al; 2017). To support chemical transport modeling and provide scientific basis for policy-making, several emission inventories covering China have been developed (Streets et al., 2003; Ohara et al., 2007; Zhang et al.,2007,
2009; Lu et al., 2010, 2011; Kurokawa et al., 2013; Klimont et al., 2009, 2013; Wang et al., 2014; Li et al., 2017; EDGAR v4.2 (available at http://edgar.jrc.ec.europa.eu)).

Bottom-up emissions are estimated through comprehensive parameterization of fuel consumption, industrial production, emission factors and mitigation measures, and spatially allocated to satisfy the chemical and climate model requirements. Uncertainties of emissions have been qualitatively illustrated (e.g., Granier et al., 2011; Saikawa et al., 2017) or
quantitatively analyzed (Streets et al., 2003; Zhao et al., 2011; Guan et al., 2012; Hong et al., 2016), inferring significant gaps in activity statistics and control measures' assumptions in emission inventories developed for different spatial scales (i.e. global, regional, or city-scale, Zhao et al., 2015).

Extensive comparisons of emission inventories have been conducted to illustrate the impacts of variable emissions on the model simulation results (e.g. Saikawa et al., 2017; Zhou et al., 2017). Although they provide important indications on the
extent of discrepancies, there are still gaps for applying the comparison results to improve the inventory accuracy:

(1)   Comparisons have been conducted for the total anthropogenic sources, instead of by sectors/subsectors/sources. Inconsistency of source categories included in inventory models were not overviewed or analyzed;

(2)   Few studies go into the comparisons on specific parameter level because the technology-based framework for each inventory was not publicly available;

(3) Top-down and bottom-up comparisons have not been comprehensively combined to infer the potential uncertain parameters for all key sectors.

To further improve the accuracy of emission estimation, we compared and evaluated the global ECLIPSE (Evaluating the Climate and Air Quality Impacts of Short-Lived Pollutants, Klimont et al., 2017) and MIX Asian (Li et al., 2017) inventories due to the following reasons: a) Up to the time of manuscript preparation, ECLIPSE and MIX are the only publicly
accessible gridded emissions dataset which include both $SO_2$ and $NO_x$ covering China for the period of 2005 and 2010; b)

Both inventories have been widely applied in atmospheric modeling and policy discussions (e.g., Stohl et al., 2015; Duan et al., 2016; Galmarini et al., 2017; Rao et al., 2017); c) The technology-based framework and compiling parameters by source categories are obtained for ECLIPSE and MIX through international collaboration, which is not accessible for other inventories over China. The methods and data were extensively described by a series of paper (Liu et al., 2015; Zheng et al., 2014; Klimont et al., 2017; Li et al., 2017), supporting us for explicit comparisons and analyses; d) ECLIPSE (GAINS model, Greenhouse gas-Air pollution Interactions and Synergies model, Amann et al., 2011) can be representative of the state-of-science global emission inventory covering China, and MIX (MEIC model, Multi-resolution Emission Inventory for China, available at www.meicmodel.org) as the regional inventory compiled with advanced methods and local data. The methods, parameters and assumptions of GAINS and MEIC are always referred to by inventory developers (e.g., Lu et al., 2010; Fu et al., 2013; Kurokawa et al., 2013; Zhao et al., 2013). The comparisons and validations are important to improve the accuracy of gridded emissions and model performance over China.

Another motivation of this work is to discuss the "fitness" of current developed inventories (specifically ECLIPSE and MIX) and modeling work done with them for policy relevant discussion. The inventories and the relevant modeling work is playing increasingly important role for policy discussion in Europe and most recently more and more in Asia at different scales. However, there is no systematic and officially approved methods and inventories but a variety of scientific products. While a lot of effort has been made to validate emission estimates with measurements, higher source and spatial resolution of inventories and projections will serve also discussion about the how to shape future policies to reduce impact of air pollution. In this work, we compared the ECLIPSE and MIX emissions over China at a detailed activity-source level. What we focused on in this manuscript is the bottom-up comparison detailed to specific parameter contributing to the differences between the two widely used gridded emission inventories (ECLIPSE and MIX), combined with top-down validations from the satellite observations. We compared the activity rates and emission factors derived from several key parameters for the largest sources for each sector/subsector. Discrepancies in the used methodologies, data sources, technology penetration assumptions, and spatial emission patterns are discussed and illustrated. Furthermore, we combined the bottom-up comparisons with top-down evaluations based on observations of the OMI (Ozone Monitoring Instrument) aboard the Aura satellite for $NO_x$ (Levelt et al., 2006). To our knowledge, it's the first emission inventory assessment work where parameter-level comparison and remote sensing evaluations are combined. OMI data provide essential constraints on emission estimates, spatial distributions and trends (Wang et al., 2012; Liu et al., 2016). In recent work described by Geng et al. (2017), OMI $NO_2$ columns were applied to analyze and evaluate the sensitivities of spatial proxies used in emission gridding process.

Methodology and data used are summarized in Section 2. Bottom-up comparisons of emissions are illustrated by decomposing the elements of inventory development in Section 3.1. Section 3.2 presents the evaluations and constraints from satellite perspective. Summary of key reasons leading to emission discrepancies are provided in Section 3.3. Finally, section 4 gives the concluding remarks.

## 2    Methodology and data

### 2.1    The ECLIPSE and MIX emission inventory

Spatially specific emission inventories of air pollutants and greenhouse gases are among key inputs for chemical transport models (CTMs) and climate models. ECLIPSE (Klimont et al., 2017) and MIX (Li et al., 2017) emission inventories have been applied in numerous modeling activities at a global (Stohl et al., 2015) and regional level, within the ECLIPSE and MICS-Asia (Model Inter-Comparison Study for Asia) Phase III project, respectively. In general, both inventories use a dynamic technology-based methodology to estimate anthropogenic emissions by multiplying activity rates with technology-specific emission factors for each source by administrative units (province / county) (Klimont et al., 2017; Li et al., 2017). Then, spatial proxies are used to distribute emission estimates by province/county to grids to satisfy the needs of model simulation. The key features of both inventories are listed in Table 1.

The **ECLIPSE** dataset is a global emission inventory for the period of 1990 to 2010 extended by projections to 2050 in five-year intervals with monthly variations, developed with the GAINS model (Amann et al., 2011). Primary sources of activity data are International Energy Agency (IEA, 2012) for fuel use and UN Food and Agriculture Organization for agriculture (FAO, http://www.fao.org/faostat/en/#home). GAINS distinguishes 172 regions, including provinces for China, for which regionally specific emission factors and technology distributions are assumed.

Emissions are distributed to grids at specific resolution (0.5°×0.5° for ECLIPSE, longitude×latitude) based on the percentages of spatial proxies located in grids by source category using GIS (Geographic Information System) techniques. For ECLIPSE, several layers were developed as spatial proxies in line with those used in the Representative Concentration Pathways (RCP) (Lamarque et al., 2010), i.e., locations of energy and manufacturing facilities, road networks, shipping routes, human and animal population density and agricultural land use. Spatial proxies were further developed within the Global Energy Assessment project (GEA project, Riahi et al., 2012), including improved population distribution, flaring in oil and gas production, smelters, and power plants for which provincial emission layers of MEIC (Multi-resolution Emission Inventory for China) were used. Spatial proxies for both ECLIPSE and MIX are summarized in Table S1.

In this work, we use gridded ECLIPSE v5a dataset (current legislation, CLE, available at http://www.iiasa.ac.at/web/home/research/researchPrograms/air/ECLIPSEv5a.html) for 2005 and 2010 in China for all anthropogenic sources excluding the international shipping and aviation to keep source consistency in comparison to MIX. We developed two sensitivity cases of the ECLIPSE emissions by changing the emission estimates or spatial proxies, to study the effect of inventory parameterization on model accuracy, as described in Sect. 3.2.1.

**MIX** was developed for 2008 and 2010 (including monthly variation) by combining the up-to-date regional inventories. For China, the monthly MEIC dataset (available at: www.meicmodel.org) and PKU-NH$_3$ (only for NH$_3$) inventory are used (Li et al., 2017). The MEIC model calculates and updates emissions for over 700 anthropogenic sources dynamically and delivers the dataset online. Activity rates are derived from local provincial statistics in China and emission factors are

derived from the best available local measurements and recent peer reviewed data for China. Power plants are treated as point sources with emissions estimated by fuel type considering actual combustion technology and installed control measures such as FGD (Flue-gas Desulfurization Devices); this information is derived from CPED (China coal-fired Power plant Emissions Database) as described by Liu et al. (2015). Following the methodology of Zheng et al. (2014), emissions of the transport sector in MEIC are estimated at county level based on comprehensive parameterization of vehicle ownership, fuel consumption, temporal evolution of emission factors, and implementation of new environmental standards. VOCs are speciated to more than 1000 species and lumped to GEOS-Chem configured mechanism based on source-specific composite profiles and mapping tables in Li et al. (2014).

Monthly gridded emissions of MEIC are generated by applying source-based spatial and temporal profiles (Li et al., 2017). Provincial emissions of MEIC are firstly distributed to county, then further distributed to grids. The former process was based on statistics by county (i.e., GDP (Gross Domestic Product), Industrial GDP, total population, urban population, rural population, agricultural activity, vehicle population), and the latter was based on gridded maps as spatial proxies (i.e., population density map, road network). For power plants, locations were determined using Google Earth following the unit-based methodology. Gridded emission product of MEIC v1.1 at a resolution of $0.25° \times 0.25°$ were integrated into MIX. In this work, we updated China's emissions with MEIC v1.2 and extended the MIX emissions back to 2005 following the same methodology.

## 2.2  GEOS-Chem

GEOS-Chem is an open-access global 3-D CTM widely used by about 100 research groups worldwide. The model is driven by the GEOS (Goddard Earth Observing System) meteorological dataset and includes complete $NO_x$-$O_x$-HC-aerosol chemistry ("full chemistry"), covering over 80 species and more than 300 chemical reactions (Bey et al., 2001; Park et al., 2004).

In this work, the Asian-nested grid GEOS-Chem model v9-01-03 driven by GEOS-5 was used to simulate the $NO_2$ maps with different emission inventories (Chen et al., 2009). Anthropogenic emissions for Asia were replaced with MIX and ECLIPSE variants described in Sect. 3.2.1. The model has a horizontal resolution of $0.667° \times 0.5°$ (lon $\times$ lat) covering Asia and 47 vertical layers. A non-local scheme was applied in mixing within the planetary boundary layer (Lin and McElroy, 2010). Global concentrations at $2.5° \times 2°$ (lon $\times$ lat) were simulated to provide time-varying boundary conditions to the target region. One-month spin-up was conducted to reduce the effect of initial conditions. To compare with the OMI observations consistently, we averaged the daily modeled vertical columns at 13:00-15:00 local time and resampled the model at grids that have OMI data.

## 2.3  Top-down emission inventory

We developed the top-down emission inventories for $SO_2$ and $NO_x$ based on the OMI/Aura L2 swath data. For $SO_2$, we obtained the Planetary Boundary Layer (PBL) $SO_2$ total vertical columns from GES DISC (Goddard Earth Sciences Data

and Information Services Center) of NASA (Li et al., 2006). The mass-balance method was used to interpret the top-down anthropogenic emissions from total columns (Martin et al., 2003; Lee et al., 2011). For $NO_x$, we used the tropospheric slant $NO_2$ columns data of DOMINO v2 (Dutch OMI $NO_2$ version 2) product accessed from the TEMIS website (Tropospheric Emission Monitoring Internet Service, http://www.temis.nl/) (Boersma et al., 2011). Slant columns were converted to vertical columns using the air mass factor (AMF), which is sensitive to the $NO_2$ vertical profile (Palmer et al., 2001; Lamsal et al., 2010). We revised the AMF by replacing the a priori vertical profiles with the modeled ones to reduce the bias in comparison following the methodology of Lamsal et al. (2010). To reduce the retrieval uncertainties, we excluded the OMI pixels at a solar zenith angle $\geq 78°$, cloud radiance fraction $> 30\%$, surface albedo $\geq 0.3$ or affected by row anomaly (http://projects.knmi.nl/omi/research/product/rowanomaly-background.php). Large pixels near the swath edges (10 pixels on each side) are also rejected in spatial averaging. Furthermore, daily data of $SO_2$ and $NO_2$ vertical column density were developed after allocating the OMI pixels to model grids ($0.667° \times 0.5°$) based on area weights.

Top-down $NO_x$ emission were developed following the Finite Difference Mass Balance (FDMB) methodology (Lamsal et al., 2011; Cooper et al., 2017). We used the summer data to develop the top-down emissions because of the stronger relationship between local emissions and grid columns. The smearing length is around 50 km over China in summer (assuming wind speed 5m/s, $NO_2$ lifetime 3h), comparable to the model grid size, implying weak effects upon the inversion of horizontal mass transport between grids. Compared to the basic mass-balance method described in Martin et al. (2003), the FDMB method reduces the errors from nonlinearity of $NO_x$-OH-$O_3$ chemistry (Gu et al., 2016; Cooper et al., 2017). A unitless scaling factor $\beta$ was introduced to represent the sensitivity of fractional modeled $NO_2$ columns to the fractional anthropogenic emission changes for each grid. We applied 15% perturbation to emissions, simulate the $NO_2$ column changes, and calculate $\beta$ following Eq. (1) (Lamsal et al., 2011; Cooper et al., 2017).

$$\frac{\Delta E}{E} = \beta \frac{\Delta \Omega}{\Omega}, \tag{1}$$

where $E$ represents the total $NO_x$ emissions, $\Omega$ represents the local $NO_2$ column. $\Delta E$ is the emission changes of anthropogenic sources, and $\Delta \Omega$ is the column changes under perturbation.

The top-down emissions were further determined based on Eq. (2).

$$E_t = E_a \left(1 + \frac{\Omega_t - \Omega_a}{\Omega_a}\beta\right), \tag{2}$$

where $E_t$ and $E_a$ represents the top-down and priori emissions, respectively. $\Omega_t$ is the OMI retrieved column. $\Omega_a$ is the modeled column of GEOS-Chem.

Following Cooper et al. (2017), we limited $\beta$ within 0.1-10, to avoid biases in regions with negligible low anthropogenic emissions or columns. The absolute error in the retrieved $NO_2$ columns is estimated at $1 \times 10^{15}$ molecules/cm$^2$ (Martin et al., 2003). We filtered out the monthly averaged retrieved columns based on this criterion, and further developed the top-down

emissions for each simulation case. Finally, summer averaged top-down emissions were developed and applied in the evaluations of this work.

## 3 Results and discussion

### 3.1 Comparisons of ECLIPSE and MIX

Following the framework of gridded emission inventory development, we conducted parameter-level comparisons between ECLIPSE and MIX, and quantified the reasons causing the emission differences for each sector. Starting from the emission comparisons for the whole China in Sect. 3.1.1, we further compared emissions by province in Sect. 3.1.2 and gridded emissions in Sect. 3.1.3.

As shown in Table 1, the activity rates were assigned independently by two inventories. As a global emission inventory,
ECLIPSE mainly relies on international statistics of IEA. Differently, MIX obtains the official statistics of energy consumption and industrial output from NBS (National Bureau of Statistics) or MEP (Ministry of Environmental Protection) of China. We can expect high independency for the determination of emission factors between ECLIPSE (GAINS model) and MIX (MEIC model). As two independently developed inventory model, the source classification, technology, removal efficiencies of control facilities in GAINS and MEIC are expected to be different although they both refer to up-to-date
measurements and peer-reviewed data. Different methods were developed in two inventory models for specific sectors, including power plants, transportation, and agriculture. For power plants, the spatial proxies were essentially consistent between ECLIPSE and MIX. For other sectors, emissions were gridded independently by two emission inventories (see Table S1).

### 3.1.1 China's emissions by sectors

Although comprehensive dataset on fuel consumption and products yield, surveys on techniques penetration, and measurements of emission factors are incorporated in the inventories, there are several additional assumptions made to characterize some sources for which information is either incomplete or missing. For ECLIPSE and MIX, assumptions are made independently and data sources are often different. Particularly, MEIC developed high-resolution emissions based on unit-based information for the power sector and county-level emissions for transportation.

Figure 1 shows the comparisons of China's emission estimates in 2005 and 2010 between two inventories for four key sectors: power, industry, residential, and transportation. For 2010, ECLIPSE estimates about 28Tg $SO_2$ and 22Tg $NO_x$ (expressed in Tg-$NO_2$ hereafter); 1% and 16% less than MIX, respectively. On a sector level, 40% difference is found for power plants (higher in ECLIPSE), 24% for the industry sector (lower in ECLIPSE) for $SO_2$, and 15%~21% in power and transportation for $NO_x$ (lower in ECLIPSE). It should be noted that heating plants are distributed in the "power" and
"industry" sectors in ECLIPSE, while aggregated into "industry" and "residential" based on the plant type of fuel combusted

in MIX. Re-distributing the heating emissions by aggregating the heating emissions from the "industry" and "residential" sectors to the "power" sector in MIX will reduce the differences to about 11% (higher in ECLIPSE) in the power sector, 17% (lower in ECLIPSE) in industry, while increase difference in the residential sector from about 10% to 32% for $SO_2$, and broaden the differences in the power sector to around 30% for $NO_x$ (ECLIPSE lower).

As shown in Fig. 1, emission trends from 2005 to 2010 are similar in two inventories, indicating analogous assumptions of technology evolution driven by economic growth and implemented air quality policies in ECLIPSE and MIX. In general, MIX estimates larger changes by sectors in the analyzed period. Specifically, for power plants, MIX estimates a decline for $SO_2$ by 54%, comparable to the 45% reduction in ECLIPSE, while for $NO_x$ both models calculate about 10% increase. MIX estimates slightly larger increasing trends for industrial emissions. For $NO_x$ emissions from transport, ECLIPSE calculates
lower overall emissions but higher growth; 26%, compared to 15% in MIX.

The fuel consumptions of MIX and ECLIPSE among different sectors in 2010 are presented in Table S2. Owing to different source-structure in each of the models, there are sometimes significant discrepancies for specific sectors. For example, for coal, the total consumption is relatively consistent, within 10% on mass basis, while in road transport sector MIX has 28% higher diesel fuel use. More details along with discussion of emissions and implied emission factors are provided below.

**Coal-fired power plants**

For power plants, coal combustion contributes more than 95% of $SO_2$ and $NO_x$ emissions. Activity rates, assumed heating values, capital sizes, emissions, and key parameters for determining emission factors for coal-fired power plants are listed and compared in Table 2.

The coal consumption of ECLIPSE is 11% ~ 14% higher than MIX in 2005 and 2010 (mass based) due to the differences in
energy statistics and included sources. As a global emission inventory model, ECLIPSE (GAINS model) relies on the energy statistics from IEA (http://www.iea.org/), consistent with the national Energy Balance Sheets provided by the NBS of China (Hong et al., 2016), and also includes district heating plants. In MIX (MEIC model), coal consumption in power plants is derived from CPED, which contains the detailed fuel consumption rates, fuel quality, combustion and control technology of over 7600 power generating units in China (Liu et al., 2015). It should be noted that the heating values of coal in China
declined between 2005 and 2010, based on the CPED database (Liu et al., 2015).

ECLIPSE's $SO_2$ emissions are 37% and 18% higher than MIX in 2010 and 2005, respectively. In 2010, the implied $SO_2$ emission factor is determined as 6.1g/kg, 24% higher than 4.9g/kg estimated by MIX. As shown in Table 2, raw emission factor, FGD application rate and removal efficiency all contribute to this discrepancy. $SO_2$ raw emission factor of ECLIPSE is 19% higher than MIX, due to different coal quality assumed in two inventory models. MIX assumes higher FGD
penetration than ECLIPSE (87.0% vs. 65.4%) while lower removal efficiency (80% vs. 95%). From 2005 to 2010, the emission discrepancy grew larger because of the decrease of sulfur content and sharply increasing application rates of FGD.

For NO$_x$, the emission estimates of power plants are similar between ECLIPSE and MIX: 7% difference in 2005 and 15% in 2010. Compared to MIX, the lower emission estimates of ECLIPSE are primarily due to emission factors, which are about 20% lower and can be explained by three factors:

*a) Fuel distributions between large and small units; in MIX, about* 89% of coal are consumed by large or medium units (> 100MW) in 2010, compared to 95% in ECLIPSE, reflecting different interpretation of mitigation strategies during 11$^{th}$ Five-Year Plan of China.

*b) Raw emission factors by technologies;* The unabated emission factor of NO$_x$ for existing large power plants differs within 5%, while for plants with LNB (low-NO$_x$ burners), emission factors are 33% lower in ECLIPSE. Furthermore, compared to MIX, ECLIPSE used 21% lower emission factors for small plants, and 26% lower for newly built plants.

*c) Application rates of technologies;* MIX assumes that in 2010 81% of power plants are equipped with the LNB techniques while only 30% application rate is considered in ECLIPSE. The impact of this difference is partly offset by higher efficiencies of LNB assumed in the latter. Neither model assumed implementation of selective catalytic reduction (SCR) installations.

**Industry**

Comparison of industrial emissions is most challenging since this sector includes multitude of sources with greatly varying emission characteristics and different representation in the investigated inventories. Overall, ECLIPSE calculates lower emissions, i.e., 24% and 13% for SO$_2$ and NO$_x$ in 2010 and 22% and 0.1% in 2005, respectively. We compare the parameters of the main contributing industrial sources below keeping the source classification differences in mind.

*Coal-fired industrial boilers;* MIX estimates about 10.4Tg SO$_2$ and 4.3Tg NO$_x$ emitted from combustion in industrial boilers, nearly 123% and 71% more than ECLIPSE. Including fuel use in fuel conversion and transformation sector in ECLIPSE slightly reduces the discrepancy to 101% for SO$_2$ and 70% for NO$_x$. While coal consumption of MIX is 29% larger than ECLIPSE, the key factor contributing to difference are varying emission factors.

*Production of cement and brick;* Cement production is among major industrial sources, contributing more than 26% industrial emissions. Both inventories use the same cement production rates, while ECLIPSE applies higher emission factors leading to 13% and 29% higher SO$_2$ and NO$_x$ emissions than MIX for 2010. For brick production, the ECLIPSE emission estimates of SO$_2$ and NO$_x$ are more than three times higher with difference in production rates of only 25%. This sector, however, is very uncertain, as information about fuel use is poorly known and actual emission factors are missing.

*Other sources;* One of major reasons for discrepancy is due to oil combustion in industrial sector where ECLIPSE assumptions indicate about 36% lower use than MIX, resulting in the 30% differences for SO$_2$ and NO$_x$ emissions. Another systematic issue is allocation of emissions between furnaces and production process where for sectors like pulp and paper, non-ferrous metals production, sinter and lime production, etc., different approaches are used in GAINS (ECLIPSE) and MEIC (MIX) models.

**Residential**

Residential combustion contributes around 10%~15% of $SO_2$ and 6% of $NO_x$ emissions in the considered period. The emission estimates are comparable between two inventories, for example for 2010, 4.15Tg $SO_2$, 1.73Tg $NO_x$ in ECLIPSE, and 4.58Tg $SO_2$, 1.38Tg $NO_x$ in MIX. Both models use nearly identical coal consumptions: about 302Tg (ECLIPSE) and 306Tg (MIX), indicating the consistent statistics from provincial energy balance table and the national ones for fuel consumed in residential boilers / stoves.

**Transportation**

Transportation sector contributes more than 25% to the total $NO_x$ emissions; negligible for $SO_2$ emissions. In ECLIPSE, high-emitters representing old and poorly maintained vehicle fleet, shares out 12% of the total transport emissions. ECLIPSE estimates 5.50 Tg $NO_x$ emissions in 2010 (4.86 Tg as shown in Table 3 and 0.64 Tg from high-emitters), 21% less than MIX. While ECLIPSE inventory includes province specific fleet characteristics (Klimont et al., 2017), MIX emissions were developed at a county level by modeling the vehicle stock following the Gompertz function, technology distributions in accordance with emission standards, and emission factors using IVE (international vehicle emission) model, as documented by Zheng et al. (2014).

Table 3 compares the fuel consumption, emission estimates and net emission factors among various vehicle types in two inventories for 2005 and 2010. Parameters for 2005 show similar difference ratio with those in 2010. Assumptions for diesel combustion sources (on-road and off-road) are the main contributor to emission discrepancies. In 2010, diesel emissions of ECLIPSE are over 50% lower than MIX estimates.

*Gasoline;* There is only 3% difference in total gasoline use in the transport sector between ECLIPSE and MIX. However, emission estimates are significantly different, especially for light duty vehicles, which dominate the total. The consistency in the total gasoline consumption between ECLIPSE and MIX is attributed to the consistency in statistics. As shown in Table 3, the gasoline consumptions by vehicle types show large differences between ECLIPSE and MIX, indicating the different vehicle fleet assumptions in two inventory models. Detailed data is not known and each of the inventories (or research groups developing them) relied on own assumptions about fuel consumption per vehicle, mileage travelled, and combined those with the available data on the number of vehicles, their sales and retirement rate. Owing to the above reasons, the results can differ significantly. Light duty vehicles are the largest gasoline consumer (> 77%) in both inventories, with 18% higher gasoline consumption estimated in ECLIPSE than those of MIX in 2010. Accordingly, ECLIPSE estimates less gasoline consumed in high duty vehicles (74%) and motorcycles (32%) than MIX. These differences reduced from 2005 to 2010. Emission estimates of HDV-G (high duty vehicles) and MC (motorcycles) also show large differences between two inventories. For HDV-G, ECLIPSE estimates lower emissions than MIX (66% in 2010), as a result of less fuel consumption while higher emission factors in ECLIPSE. For MC, emissions of ECLIPSE are 64% lower than MIX, contributed by both fuel consumption and emission factors. It appears that the assumptions about penetration and performance of vehicles with

specific emission standards vary between the models since the fleet average emission factor in ECLIPSE is 69% higher than that of MIX, i.e., 9.3 g/kg vs. 5.5 g/kg.

*Diesel;* As shown in Table 3, the significant diesel emission discrepancies can be primarily attributed to the differences in fuel consumption. Compared to MIX, ECLIPSE has 22% lower diesel use for on-road vehicles and 39% lower for off-road engines. While applied emission factors are comparable for most categories, there is large discrepancy for light duty vehicles where MIX value is four times larger than ECLIPSE. One possible explanation is that there might be an issue with assumptions about fuel efficiency that was applied when converting the native MEIC values which are kilometers driven for activity and gram per kilometer for emission factors (Zheng et al., 2014)

### 3.1.2 Provincial emission estimates

Provincial emissions were developed by different methodologies for two inventories (see Sect. 2.1). The provincial emission discrepancies between two inventories are attributed primarily to two factors: (i) the differences in activities, emission factors and policy implementation assumptions at the national level (as discussed in previous section) and (ii) distribution of activities among the provinces – see section 2.1 for principal data sources for the latter.

Figure 2 compares emissions and relative difference in fossil fuel consumption by province in ECLIPSE and MIX in 2010. The differences in provincial $SO_2$ emissions are relatively large for a number of provinces, especially when compared to the fluctuations of coal consumption. This indicates significant differences in provincial emission factors which is mainly because of varying assumptions on application and efficiency of abatement measures but also different allocation of coal use between power and industry since emission standards for these sectors are different. MIX estimates lower emissions mainly for eastern China, including Shandong, Hebei, Henan, Jiangsu, Sichuan, Zhejiang, and Anhui.

For $NO_x$, ECLIPSE estimates are systematically lower than MIX for most provinces. For twenty provinces, mainly located in northern and central China, such as Hebei, Shanxi, Inner Mongolia, Liaoning, Jilin, Heilongjiang and Shandong, ECLIPSE emissions are lower by over 20%. For Beijing, on the other hand, ECLIPSE emissions are 41% higher than MIX driven by larger estimate for power plants (+36%) and transport (+114%). In general, assumptions about diesel consumption in transport vary significantly between inventories highlighting the need for further validation of the regional fuel statistics. It is important to note that, as in many other countries, the national and regional energy use statistics contain limited information about diesel fuel use in trucks, and non-road engines used in industry where fuels are allocated to industry rather than transport.

The sectorial distributions of emissions by province are generally consistent between two inventories, as presented in Fig. S1. For $SO_2$, the emission fractions of power plants in MIX are lower, and industrial fractions are overall higher than those of ECLIPSE, due to the differences in source classification and emission factors. The distribution patterns of $NO_x$ provincial emissions show relatively good consistency (within 30% difference on sector level) between two inventories.

### 3.1.3 Gridded emissions

Gridded emissions are direct inputs for atmospheric chemistry models and climate models. We compared ECLIPSE and MIX gridded emissions by analyzing three components: emissions by grids at $0.5° \times 0.5°$ resolution in 2010, the spatial proxies used in gridding process, and gridded emission trends from 2005 to 2010.

*Gridded emissions;* Figure 3 compares the gridded emissions between ECLIPSE and MIX for $SO_2$ and $NO_x$. MIX emissions were aggregated from $0.25°$ to $0.5°$ to be comparable with ECLIPSE. The discrepancies in spatial distribution of gridded emissions are in line with provincial emission differences discussed earlier. Grids located in eastern parts of China show higher $SO_2$ emissions in ECLIPSE, compared to MIX. $NO_x$ emissions of ECLIPSE are overall lower than MIX, except Beijing and Guangzhou. Correlations between two gridded emissions are quite good at $0.5° \times 0.5°$ grids (slope = 0.83, R=0.9 for $SO_2$, slope = 0.79, R=0.98 for $NO_x$).

Sectorial emissions show distinct spatial characteristics (Fig. 3(b)). Comparisons of industrial and residential sectors show clear administrative boundaries as these are typically distributed from provincial emissions using population-based proxies. Since power plants are treated as point sources, emissions differ in grids over the entire country, higher for $SO_2$ and lower for $NO_x$ in ECLIPSE. Signals of large cities are observed in the comparison for transportation sector because emissions are gridded based on road network or population distribution.

The differences of gridded emissions illustrated in Fig. 3 are attributed to the discrepancies in emission estimates nationwide and by provinces (Sect. 3.1.1, Sect. 3.1.2), and also method and data of emission spatial allocations (see Sect. 2). For power plants which were treated as point sources, emissions are gridded based on the locations verified by Google Earth (Liu et al., 2015), consistent between ECLIPSE and MIX. For other sectors, ECLIPSE gridded the provincial emissions according to the source-specific layers, and MEIC used two-step allocation method (province to county, county to grid). The data sources of spatial proxies also differ between two inventories (see Table S1). We further compared the spatial proxies by sectors in Figure 4.

*Spatial proxy;* Spatial proxies can play key roles in evaluating the accuracy of emission inventory and CTM simulation (Geng et al., 2017; Zheng et al., 2017). Proxies used in ECLIPSE and MIX emission gridding process are summarized in Sect. 2. Source-specific layers were developed as spatial proxies by ECLIPSE, among which, MEIC emissions were taken to distribute emissions for power plants (Klimont et al., 2017). For industry and residential sector, emissions are distributed mainly based on population data. Road networks and population are used as proxy for transportation emissions. The spatial proxies used in MIX (MEIC) have been summarized in several papers (Geng et al., 2017; Li et al. 2017), showing that local proxies are used in gridding process. MIX (MEIC) uses Google Earth in verifying the locations for each power plant. As described in Zheng et al. (2014), for the transport sector, the China Digital Road-network Map is used for emission distribution. Other proxies including the total population map (for some industrial sources), urban population map (for industrial heating, residential coal burning, etc.) and rural population map (for residential biofuel burning), are in general consistent with global inventory (Geng et al., 2017).

In this work, we calculated the distribution ratios, reflecting the spatial proxies used, by dividing the emissions for each grid by the provincial emissions for each sector. The distribution ratios between ECLIPSE and MIX in 2010 are shown in Fig. 4. Excellent correlations (slope $\geq 0.87$, $R \geq 0.94$) are observed for all sectors, which is reasonable because similar proxy dataset were used in two inventories, as illustrated above. The differences for specific sectors (e.g., residential with slope of 0.87) are slightly higher than others, mainly due to the different population dataset used for emission allocation of relevant sources in ECLIPSE and MIX.

*Emission trend (2005 ~ 2010);* Figure 5 presents the emission changes of $SO_2$ and $NO_x$ estimated by ECLIPSE and MIX for the period of 2005 to 2010. For $SO_2$, the emission trends are similar between the two inventories: sharp decrease for power plants due to the wide application of desulfurization facilities since 2006, and overall increase for industrial sources driven by economic growth and still low penetration of emission control technology, consistent with the national emission trend analyses in Sect. 3.1.1. For $NO_x$, different emission trends are estimated for transportation and consistent trends for other sectors.

In MIX, decreasing emissions for Beijing and PRD (Pearl River Delta) region are estimated, which are dominated by decline in power and transport sectors, in contrast to the increasing emission trend of ECLIPSE. The different trends of transportation emissions are attributed to the different assumptions on legislation effect on pollution control in two inventory systems. For Beijing, the differences of transportation emission trend are mainly caused by diesel vehicles. In ECLIPSE, 47% increases are estimated for diesel fueled vehicles, compared to 28% emission decreases in MIX. Fuel consumptions show large discrepancies in trend from 2005 to 2010, where +54% (ECLIPSE) compared to -20% (MIX) for high duty vehicles, and +45% (ECLIPSE) compared to +3% (MIX) for light duty vehicles. The emission factors of light duty vehicles increase by 5% in ECLIPSE, while decrease by 34% in MIX, attributed to the different assumptions on emission control effects. As a pioneer in pollution control of China, Beijing carried out Euro III standard in 2005 and Euro IV standard in 2008 for light duty vehicles. The Euro IV penetrations in 2010 in Beijing are assumed around 12% in ECLIPSE, while more than 60% in MIX, which might be too optimistic and should be verified with local surveys.

For the PRD region, gasoline and high duty diesel vehicles contribute to the different emissions trend. 22% emission growth for LDB-G (light duty gasoline buses) is estimated in ECLIPSE, compared to 12% emission reduction in MIX. For high duty diesel vehicles, trend of fuel consumption (+55% in ECLIPSE, compared to -11% in MIX) and technology distribution (21% of Euro III in 2010 for ECLIPSE, compared to >50% in MIX) are the main contributors to the difference. In summary, survey data are urgently needed to validate the fuel consumptions, effect of legislation effect and trend for diesel vehicles in pioneering regions such as Beijing and PRD.

**3.2   Evaluations from satellite perspective**

In this section, we evaluated the effect of emission inventories on the accuracy of model simulations through combing GEOS-Chem Asian-nested modeling and OMI observations (Sect. 3.2.1). Top-down emissions were developed for both $SO_2$

and NO$_x$ (Sect. 3.2.2). Due to the large uncertainties in SO$_2$ retrievals of OMI, we mainly focused on the evaluations for NO$_x$ emission estimates, spatial proxies and emission trend.

### 3.2.1 Sensitivity cases for model simulations

The main purpose of this sub-section is to evaluate the effect of gridded emissions on model performance, and figure out the effect of emission estimates and spatial distributions on model performance, using satellite observations as criterion. Therefore, we set up four sensitivity cases of modeling, ECL-case0, ECL-case1, ECL-case2 and MIX. ECL-case0 and MIX form two basic cases, which apply the ECLIPSE and MIX emissions in the simulation, respectively. ECL-case1 scales the China's emissions of ECLIPSE to the MIX's value by sectors retaining original spatial distributions. ECL-case2 re-distributes the ECLIPSE emissions over China based on the spatial grid ratios of MIX, also on sector level. The characteristics of the emission inventory used for each case are summarized in Table 4 and shown in Fig. S2. We processed each inventory into model-ready inputs through regridding new emissions, performing VOC speciation and temporal allocation. The speciation factor and monthly profiles by sectors of the MIX inventory are used for ECL-case0~case2. We re-sample the model results based on satellite observations in spatial and temporal for consistent comparison as described in Sect. 2.3.

Figure 6 compares the SO$_2$ columns simulated by four model cases and OMI SO$_2$ columns in 2010. Although OMI data tend to overestimate the concentrations due to the overlap in signals of SO$_2$ and O$_3$ during retrieval, good correlations are found between model results and satellite (R=0.633-0.667, Slope=0.842-0.863, general consistent among sensitivity cases, see Table S4), confirming the high accuracies of the priori SO$_2$ spatial emission patterns.

NO$_2$ tropospheric vertical columns modeled for each case are compared with the retrieved OMI columns (Fig. 7). Summer averaged results are shown here because of the closer connection between emissions and columns due to short NO$_x$ lifetime. As shown in Fig. 7, modeled NO$_2$ density map shows similar spatial pattern among cases, but different magnitudes. Higher NO$_2$ concentrations are observed for the ECL-case1 and MIX case, because common emission estimates of MIX are used, which are higher than ECLIPSE. Compared to OMI, all model cases underestimate the pollution in northern China and slightly overestimate the columns over central China. As illustrated in Table 4, the performance of MIX case is the best among all cases, identified with low biases (NMB = -4.72%) and better slope ratio (slope = 0.601). The results of ECL-case0 and ECL-case2 are quite similar because the differences of spatial proxies of two inventories are negligible (Sect. 3.1.3). Replacing the emission estimates of ECLIPSE with MIX improves the model performance from bias at -12.2% to -6.19% (ECL-case1 vs. ECL-case0).

### 3.2.2 Top-down emission evaluations

Satellite-based emission inventories were developed following the finite difference mass balance methodology (Cooper et al., 2017). Emissions of SO$_2$ estimated by bottom-up and top-down inventories are presented in Table S5. Both ECLIPSE and MIX correlates well with the top-down estimates (R=0.722-0.896, Slope = 0.539-0.923) in 2005 and 2010. Relatively high

negative biases are found (NMB = -51.0% ~ -29.1%) in the bottom-up inventories, may attributed to the uncertainties in the OMI retrievals for $SO_2$. Table 5 shows the $NO_x$ emission estimates and correlations between bottom-up and top-down inventories. For $NO_x$, it can be concluded that ECLIPSE and MIX are consistent with the top-down estimates over China. Summer averaged bottom-up emissions show strong correlations with the top-down ones (R > 0.87 for both inventories) in

2010. The mean biases of MIX are -21.2% (11.9 mole/s in RMSE), much lower than -39.4% of ECLIPSE (14.6 mole/s in RMSE). The slope ratio of MIX show slightly better performance than ECLIPSE (0.73 for MIX, 0.50 for ECLIPSE), but should be interpreted with caution since the slope can be dominated by several large point sources.

In spatial distribution, large discrepancies are observed between bottom-up emission inventory and the top-down ones. For $SO_2$, bottom-up inventories tend to underestimate emissions in Shandong province and several southern provinces such as

Guizhou, Jiangxi, and Fujian which may attributed to the scattered coal consumption, while overestimate emissions in the YRD region (see Figure 8). As shown in Fig. 9, both ECLIPSE and MIX underestimate the $NO_x$ emission strength in northern China, parts of YRD (Yangtze River Delta) and PRD regions, and overestimate emissions located in large cities such as Beijing, Shanghai, and Wuhan. One important reason for the latter is associated with the limitations of currently used spatial proxies. Using population or IGDP (Industry Gross Domestic Product) as spatial proxy may distribute too much

emissions to provincial capitals or economically developed cities. Through sensitivity test analyses, it's concluded that treating sources as point sources can significantly reduce the uncertainties in emission gridding process (Geng et al., 2017).

Emission changes from 2005 to 2010 were evaluated and presented in Fig. 10 for $SO_2$ and Fig. 11 for $NO_x$. The maps of $SO_2$ emission changes are consistent in spatial patterns between bottom-up and top-down inventories. Effective control measures including nationwide FGD application led to significant $SO_2$ emission decrease between 2005 and 2010, especially in

Beijing, Hebei, Shanxi, YRD, PRD and southwest provinces of China. The annual growth rate of China's emissions of $NO_x$ is highly consistently estimated by ECLIPSE, MIX and satellite-based inventories, around 4.0% annual growth in the period of 2005 to 2010 (see Table 5). The results are comparable with previous work using various inversion methodology, satellite sensor or CTMs (Gu et al., 2013; Krotkov et al., 2016; Miyazaki et al., 2017). Figure 11 show the gridded emission changes from 2005 to 2010 among different inventories for $NO_x$. Decease in parts of YRD and PRD region, and shut down of large

facilities are captured by satellite, showing general consistent map with MIX. Significantly larger growth is observed in northern China's emissions from top-down inventories, than in estimates of ECLIPSE and MIX. In Beijing, the satellite-based inventory shows relatively stable trend, different from the increasing trend of ECLIPSE or decreasing trend of MIX, indicating that assumptions about the penetration of emission reduction technology need further revision in both inventory models for large cities.

## 3.3 Discussion

### 3.3.1 Summary of key parameters contributing to emission uncertainties

We address several key factors contributing to the differences between ECLIPSE and MIX for $SO_2$ and $NO_x$ emission estimates: source classification, energy statistics, emission factors, assumptions about control technology penetration, and spatial proxies.

The source classification for heating plants, fuel conversion, and industrial boilers are differently defined between ECLIPSE (GAINS model) and MIX (MEIC model), making the interpretation of comparisons for each source more difficult and to some degree less transparent (Sect. 3.1.1). The source-structure differences are inevitable for emission inventory models designed for estimates at different spatial scales. As a global inventory model, GAINS integrates statistics from international sources (e.g. IEA, FAO). Therefore, the source-structure of GAINS is set up in accordance with the international statistics framework. Focusing on emissions on regional scale, MEIC set up calculation framework based on statistics from local agencies in China to gain higher specificity in temporal and spatial distribution (e.g. NBS, CPED). As illustrated and analyzed in Sect. 3.1.1, the differences by sectors should be interpreted with caution, especially for power and industry sectors.

The apparent emission uncertainty ratio (the ratio of the maximum emission discrepancy to the mean value using provincial energy statistics or national statistics) of $SO_2$ and $NO_x$ resulting from energy use are 30% and 16% according to Hong et al. (2016). For $SO_2$, emission uncertainties are quite sensitive to the energy use uncertainty mostly contributed by industrial coal use (Hong et al., 2016). Based on this work, diesel consumption in transport sector remains highly uncertain contributing to the emission differences for $NO_x$.

The FGD penetration rate in power plants, as well as assumed removal efficiencies, significantly affect the $SO_2$ emission estimates and trends. Similarly, for $NO_x$, application rates and abatement efficiency of LNB technology installed in power plants is significantly different in the compared models; these assumptions should be further verified and constrained. Emission factors for diverse industrial boiler types are the main contributors to the uncertainty in the industrial emissions. Assumptions about vehicle fleet, implementation of emission standards, and emission factors for various vehicle types still differ between the investigated inventory models. More in-situ measurements and local surveys are needed to reduce these uncertainties.

Spatial proxies used in emission inventories are an important factor contributing to the overall accuracy in model simulation. Integration of detailed spatial information that is often included in regional inventories like MIX should be considered as the best way to improve the resolution and spatial allocation of emissions in global products like ECLIPSE.

### 3.3.2 Uncertainty of top-down evaluation

In this work, moderate negative biases are observed in bottom-up emission inventories (-21% for MIX, -39% for ECLIPSE), compared to satellite-based ones. But the top-down evaluations are subject to uncertainties from both satellite retrievals and model simulations. The uncertainties of retrieved individual $NO_2$ column of DOMINO v2.0 product are estimated at $1.0 \times 10^{15}$ molecules cm$^2$, +25% mainly arising from the AMF calculation (Boersma et al., 2007, 2011). Because of the high aerosol loadings in eastern China, the aerosol scattering and absorption have positive or negative effects on $NO_2$ retrieval, with a mean effect of 14% (Lin et al., 2014). Negative systematic bias of 10-20% by seasons plus a random error of 30% are generated by model simulation using GEOS-Chem (Martin et al., 2003; Lin and McElroy, 2010; Lin, 2012). As suggested by Ding et al. (2017), multiple sensors can give more comprehensive and accurate constraints on the priori spatial and temporal emission estimates, which can be further applied in future work.

## 4    Concluding Remarks

We conducted parameter-level comparisons of gridded China's emissions between ECLIPSE and MIX, elucidated the effect on CTM simulations, and evaluated the inventories based on OMI observations. The work is important for inventory developers and modelers for understanding the potential uncertainties in the gridded emission inventory over China. For inventory developers, the detailed comparisons give indications on the underlying uncertainties of parameters by sources, including the source classifications, activity rates, emission factors and technology distributions. For modelers, the comparisons and validations are important to understand the effect of emissions on model performance. This work shows that our best inventories appear to be fit for evaluation of the policies at an aggregated or national level, more work is needed in specific areas in order to improve accuracy and robustness of outcomes at the finer spatial and also technology level. The main findings are:

a) In 2010, compared to MIX, the emission estimates of ECLIPSE are identical for $SO_2$, and 16% lower for $NO_x$. $SO_2$ emissions of power plants and industry sectors differ by +40% and -24% (ECLIPSE compared to MIX), attributed to the differences in source classification system, FGD penetration rates, and assumed removal efficiencies. Emission factors for diverse industrial boiler types are the main reason for the industrial emission differences. For $NO_x$, ECLIPSE estimates are lower than those of MIX for all sectors. Lower $NO_x$ emission factors for power plants, and lower diesel consumptions in the transport sector in ECLIPSE are the main reasons for the discrepancies. Application rates and abatement efficiency of plants equipped with LNB should be further verified and constrained. Assumptions about vehicle fleet, implementation of emission standards and emission factors for various vehicle types still differ between evaluated inventory models. Large uncertainties should be addressed for the diesel consumptions in current inventory models.

b) We modeled four sensitivity cases to investigate the effect of emission estimates and spatial proxies of emission inventories on model accuracy using GEOS-Chem. The model case using MIX as input show the best performance, with mean biases at -4.72% (NMB, for $NO_x$). Increasing the ECLIPSE emission estimates to MIX reduces the biases from -12.2%

to -6.19% (for $NO_x$). For ECLIPSE, changing spatial pattern to MIX does not affect the model results apparently, owing to the role of power sector (ECLIPSE uses MEIC proxy already) plays in emissions. Top-down emissions were developed based on OMI retrievals. High correlations were observed between the bottom-up and top-down $SO_2$ emissions, providing evidence of the accuracy of the spatial emission patterns in ECLIPSE and MIX. We found moderate negative biases in bottom-up emission inventories for $NO_x$ (-21% for MIX, -39% for ECLIPSE), compared to satellite-based ones.

c) Both inventories show decreasing trends for $SO_2$ and increasing trends for $NO_x$ between 2005-2010 but the spatial pattern of change differs. Signals of large power plants and of city center can be found. Trend analyses from top-down perspective indicate annual growth rate of 4% for $NO_x$; consistent with development of bottom-up emissions. A strong $NO_x$ emission increase in northern China and decrease in parts of YRD and PRD regions are captured by the satellite retrievals, similar to the MIX estimates.

*Data Availability.* ECLIPSE v5a global emissions developed based on the GAINS model can be open accessed from http://www.iiasa.ac.at/web/home/research/researchPrograms/air/ECLIPSEv5a.html. The specific parameters in GAINS are achieved from http://www.iiasa.ac.at/. The MIX inventory is publicly available from http://www.meicmodel.org/dataset-mix.html. China's emissions in MIX are obtained from MEIC v1.2, which are downloaded from http://www.meicmodel.org/index.html. The L2 swath $SO_2$ column data developed by NASA are available at https://disc.gsfc.nasa.gov/. The tropospheric $NO_2$ column data of DOMINO v2 can be accessed from www.temis.nl.

*Acknowledgements.* This work was supported by the National Key R&D program (2016YFC0201506), the National Natural Science Foundation of China (41625020), and IIASA's Young Scientists Summer Program (YSSP) sponsored by the National Natural Science Foundation of China (41611140118). We acknowledge the free use of NASA OMI $SO_2$ and DOMINO v2 product.

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

## Tables and Figures

**Table 1. Key features of ECLIPSE v5a and MIX emission inventories.**

| Item | ECLIPSE v5a | MIX |
|---|---|---|
| Year | 1990-2010 at a 5-year interval | 2005[a], 2008, 2010 |
| Domain | Global | Asia |
| Spatial resolution | 0.5°×0.5° | 0.25°×0.25° |
| Temporal resolution | Monthly | Monthly |
| Activities included for each sector | | |
| Energy / Power | Power plants (including CHP), energy production/conversion (including district heating plants), fossil fuel distribution | Power plants (including CHP) |
| Industry | Industrial combustion and processes | Industrial combustion (including industrial heating plants) and industrial processes |
| Residential | Residential combustion sources | Residential combustion sources (including residential heating plants) |
| Transportation | On-road and off-road transport sources [b] | On-road and off-road transport sources [b] |
| Agriculture | Livestock and fertilization | Livestock and fertilization |
| Data sources of activity rates | | |
| Power | International Energy Agency (IEA) | CPED (Liu et al., 2015) |
| Industry | International Energy Agency (IEA) | Provincial industrial economy statistics (NBS) |
| Residential | International Energy Agency (IEA) | Provincial energy statistics (NBS) |
| Transportation | International Energy Agency (IEA) | Provincial energy statistics (NBS); Zheng et al. (2014) |
| Agriculture | UN Food and Agriculture Organization [c] | Provincial statistics (NBS, Huang et al., 2012) |
| Emission factors and technology | GAINS model (Klimont et al., 2017) | MEIC model [d], Process-based model for $NH_3$ (Huang et al., 2012) |
| Data Access | http://www.iiasa.ac.at/web/home/research/researchPrograms/air/ECLIPSEv5a.html | http://www.meicmodel.org/dataset- mix |

[a] Developed following the same methodology of Li et al. (2017)

5    [a] International air and international shipping are not included.

[c] FAO, http://www.fao.org/faostat/en/#home.

[d] Zhang et al., 2009; Lei et al., 2011; Zheng et al., 2014; Liu et al., 2015

**Table 2. Activity rates, emissions and emission factors for $SO_2$ and $NO_x$ in power plants of China, 2005 and 2010[a].**

| Category | Sub-category | ECLIPSE | | MIX | |
|---|---|---|---|---|---|
| | | 2005 | 2010 | 2005 | 2010 |
| Activity rates | Heating value, MJ/kg | 20.7 | 20.7 | 19.0[b] | 18.8[b] |
| | Energy consumption, Tg (PJ) | 1202 (24890) | 1743 (36074) | 1055 (20084) | 1577 (29758) |
| Capacity size | < 100MW | 39.9%[c] | 5.1%[c] | 25.0% | 11.5% |
| | ≥ 100MW | 60.1% | 94.9% | 75.0% | 88.5% |
| $SO_2$ emissions and emission factors | $SO_2$ emissions, Gg | 19528 | 10645 | 16516 | 7754 |
| | Average $SO_2$ emission factor, g/kg (g/MJ) | 16.2 (0.79) | 6.11 (0.29) | 15.6 (0.82) | 4.92 (0.26) |
| | Sulfur content, % | 1.13 | 1.13 | 1.04 | 0.95 |
| | Sulfur retention in ashes, % | 0.092 | 0.092 | 0.15 | 0.15 |
| | Raw $EF_{SO2}$[d], g/kg | 22.5 | 22.5 | 20.8 | 19.0 |
| | Removal efficiency of FGD, % | 95 | 95 | 80 | 80 |
| | Application rate of FGD, % | 17.6 | 65.4 | 12.6 | 87.0 |
| $NO_x$ emissions and emission factors | $NO_x$ emissions, Gg | 6131 | 7090 | 6561 | 8302 |
| | Average $NO_x$ emission factor, g/kg (g/MJ) | 5.10 (0.25) | 4.07 (0.20) | 6.22 (0.33) | 5.27 (0.28) |
| | LNB penetration, % | 29.4 | 30.1 | 53.7 | 81.4 |
| | Unabated $EF_{nox}$, existing large PP, g/kg | 7.55 | 7.55 | 7.21 | 7.21 |
| | Unabated $EF_{nox}$, existing small PP, g/kg | 7.04 | 7.04 | 8.96 | 8.96 |
| | LNB $EF_{nox}$, existing large PP, g/kg | 3.78 | 3.78 | 5.63 | 5.63 |
| | LNB $EF_{nox}$, existing small PP, g/kg | 3.52 | 3.52 | 7.00 | 7.00 |
| | LNB $EF_{nox}$, newly built PP, g/kg | 3.11 | 3.11 | 4.21 | 4.21 |

[a] Including both raw coal and derived coal.

[b] National average.

5    [c] We interpret the defined small units (< 50MW) in ECLIPSE to (<100MW) here by assuming 1/3 units in range of (0,100MW) fall into (0, 50MW) according to Liu et al. (2015).

[d] The raw $EF_{SO2}$ is calculated following:    $EF = 2 \times sulfur\ content \times (1 - sulfur\ retention\ in\ ashes)$.

**Table 3. Comparisons of activity, emissions, and emission factors for the transport sector emission estimates of $NO_x$[a].**

| Items | Inventory-Year | HDV-G | LDV-G | MC | All gasoline on-road | HDV-D | LDV-D | All diesel on-road | Diesel off-road |
|---|---|---|---|---|---|---|---|---|---|
| Fuel consumptions, Tg | ECL-2005 | 3.63 | 35.0 | 8.57 | 47.2 | 29.7 | 12.8 | 42.5 | 23.1 |
| | ECL-2010 | 1.50 | 62.4 | 7.03 | 71.0 | 51.0 | 20.9 | 72.0 | 25.8 |
| | MIX-2005 | 13.3 | 25.1 | 8.30 | 46.7 | 59.7 | 6.11 | 65.8 | 35.2 |
| | MIX-2010 | 5.76 | 52.8 | 10.3 | 68.9 | 81.2 | 11.0 | 92.2 | 42.1 |
| $NO_x$ emissions, Gg | ECL-2005 [b] | 109 | 722 | 46 | 878 | 1570 | 193 | 1762 | 1166 |
| | ECL-2010 [b] | 27 | 582 | 34 | 643 | 2667 | 335 | 3002 | 1215 |
| | MIX-2005 | 208 | 314 | 129 | 652 | 3170 | 398 | 3568 | 1854 |
| | MIX-2010 | 79 | 292 | 92 | 463 | 3614 | 705 | 4319 | 2201 |
| Average $NO_x$ emission factors, g/kg (g/MJ)[b] | ECL-2005 [c] | 30.1 (0.70) | 20.6 (0.48) | 5.4 (0.13) | 18.6 (0.43) | 52.8 (1.22) | 15.1 (0.35) | 41.4 (0.96) | 50.4 (1.17) |
| | ECL-2010 [c] | 18.0 (0.42) | 9.3 (0.22) | 4.8 (0.11) | 9.1 (0.21) | 52.2 (1.21) | 16.0 (0.37) | 41.7 (0.97) | 47.1 (1.09) |
| | MIX-2005 | 15.6 (0.36) | 12.5 (0.29) | 15.6 (0.36) | 14.0 (0.32) | 53.1 (1.23) | 65.1 (1.51) | 54.2 (1.26) | 52.7 (1.22) |
| | MIX-2010 | 13.7 (0.32) | 5.5 (0.13) | 9.0 (0.21) | 6.7 (0.16) | 44.5 (1.03) | 64.1 (1.49) | 46.8 (1.09) | 52.3 (1.21) |

[a] HDV-G: high / medium duty buses and trucks – gasoline fueled; LDV-G: light duty buses, trucks, and passenger cars – gasoline fueled; MC: motorcycle; HDV-D: high / medium duty buses and trucks – diesel fueled; LDV-D: light duty buses and trucks – diesel fueled.

5 [b] High emitters are not included.

[c] Emission factors on mass base are converted to energy base with heating value of 43.1 MJ/kg for gasoline and diesel.

**Table 4. Description of model simulation cases and statistics of model performance of NO$_2$, summer average in 2010.**

| Simulation cases | Emission estimates | Spatial proxies | R | Slope | NMB (%) | RMSE ($10^{15}$ mole/cm$^2$) |
|---|---|---|---|---|---|---|
| ECL-case0 | EM-ECL[a] | SP-ECL[b] | 0.814 | 0.476 | -12.2 | 1.19 |
| ECL-case1 | EM-MIX[c] | SP-ECL | 0.818 | 0.559 | -6.19 | 1.15 |
| ECL-case2 | EM-ECL | SP-MIX[d] | 0.824 | 0.474 | -12.1 | 1.18 |
| MIX | EM-MIX | SP-MIX | 0.811 | 0.601 | -4.72 | 1.16 |

[a] ECLIPSE emission estimates by sectors in China.

[b] Spatial proxies by sectors based on ECLIPSE.

5   [c] MIX emission estimates by sectors in China.

[d] Spatial proxies by sectors based on MIX.

**Table 5. Top-down NO$_x$ emission evaluations over China [a].**

| Inventories | ECLIPSE | | MIX | |
|---|---|---|---|---|
| Year | 2005 | 2010 | 2005 | 2010 |
| Bottom-up emissions (kmole/s) | 10.4 | 12.2 | 12.5 | 15.4 |
| Top-down emissions (kmole/s) | 17.0 | 20.2 | 16.0 | 19.6 |
| R | 0.936 | 0.866 | 0.936 | 0.891 |
| Slope | 0.598 | 0.500 | 0.821 | 0.726 |
| NMB (%) | -38.6 | -39.4 | -21.8 | -21.2 |
| RMSE (mole/s) | 9.48 | 14.6 | 7.47 | 11.9 |

[a] Only contain grids that covered by OMI pixels filtered by criterion described in the text.

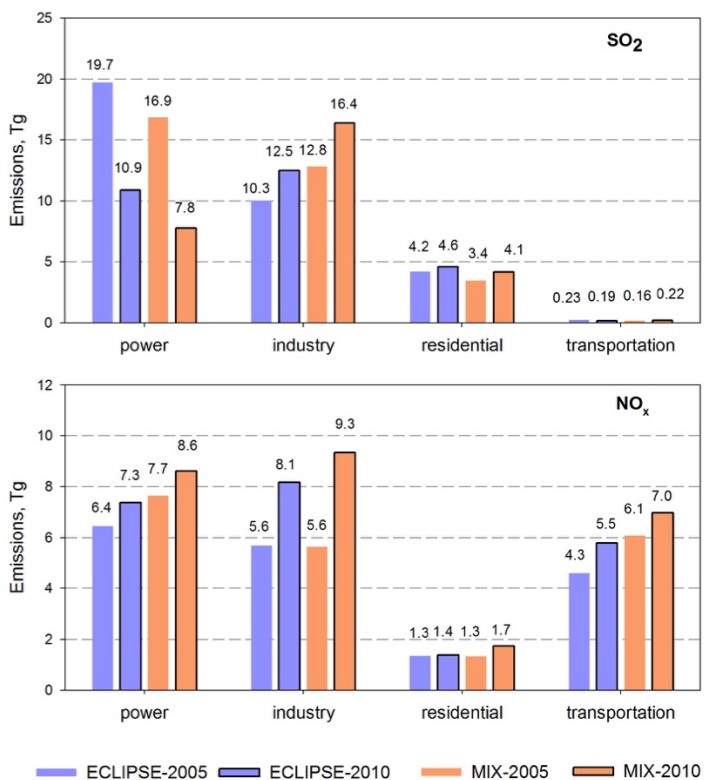

**Figure 1. Emissions of SO$_2$ and NO$_x$ in 2005 and 2010 by sectors over China. "ECL" is the abbreviation of "ECLIPSE". NO$_x$ emissions are shown in Tg-NO$_2$.**

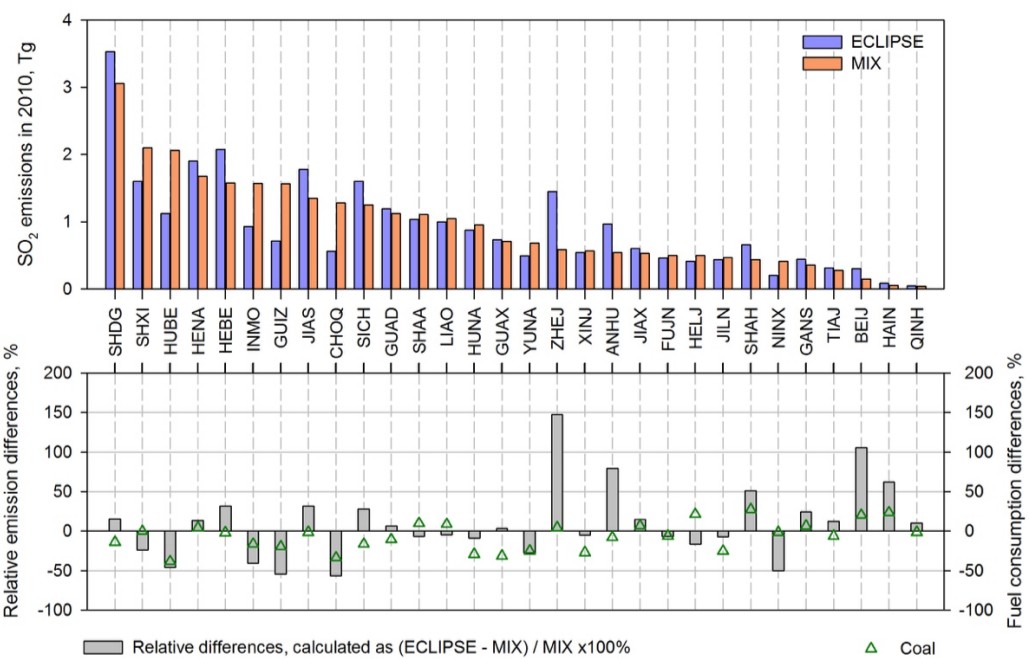

(a) SO₂

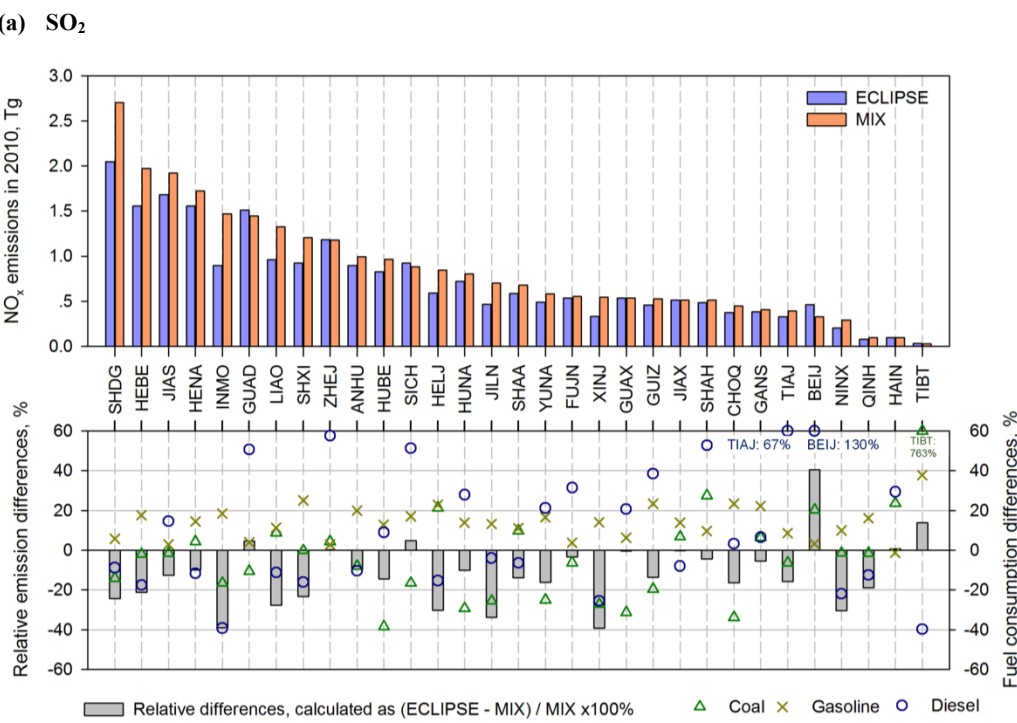

(b) NOₓ

Figure 2. Comparisons of emission estimates and fuel consumptions by provinces in China, 2010. Values out of y-axis range are labeled separately in the graphs. Abbreviations of provinces are provided in Table S3.

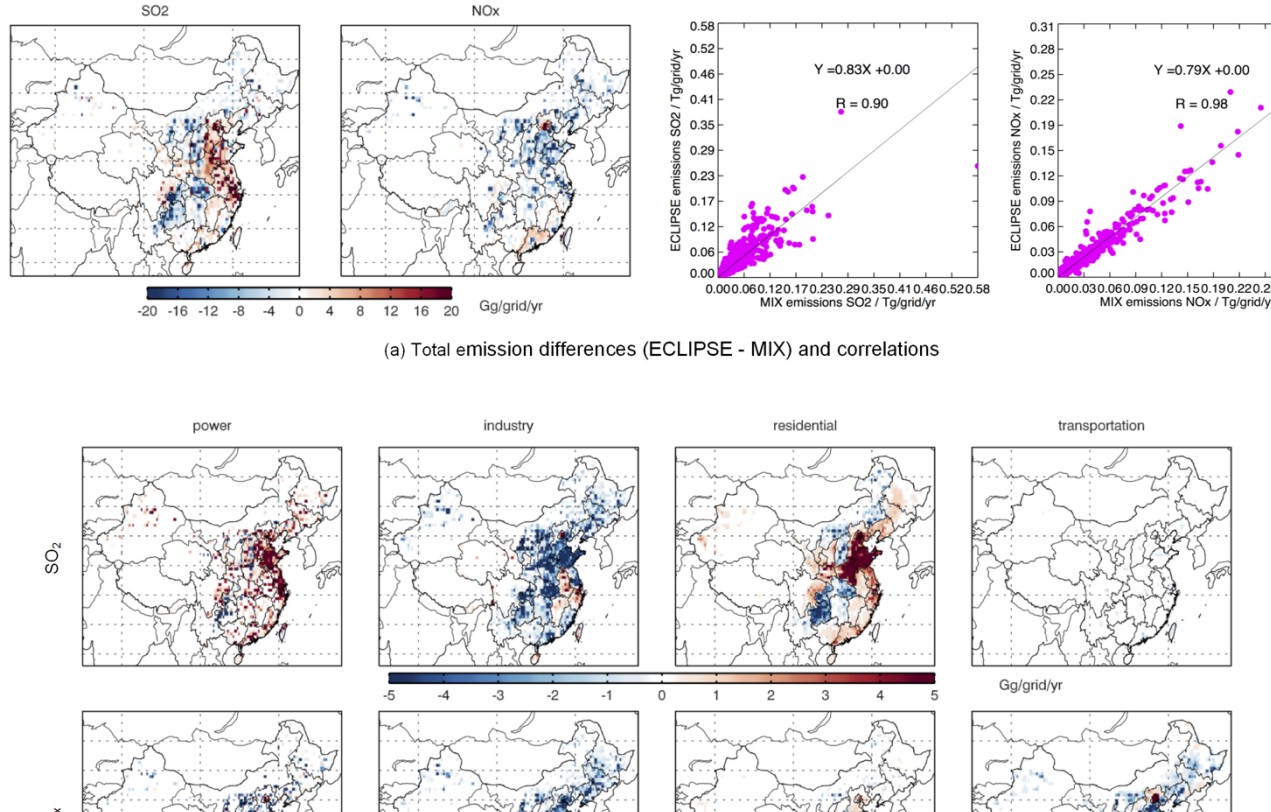

(a) Total emission differences (ECLIPSE - MIX) and correlations

(b) Emission differences by sectors (ECLIPSE – MIX)

**Figure 3. Comparisons of MIX and ECLIPSE gridded emissions, 0.5°×0.5° grids, 2010.**

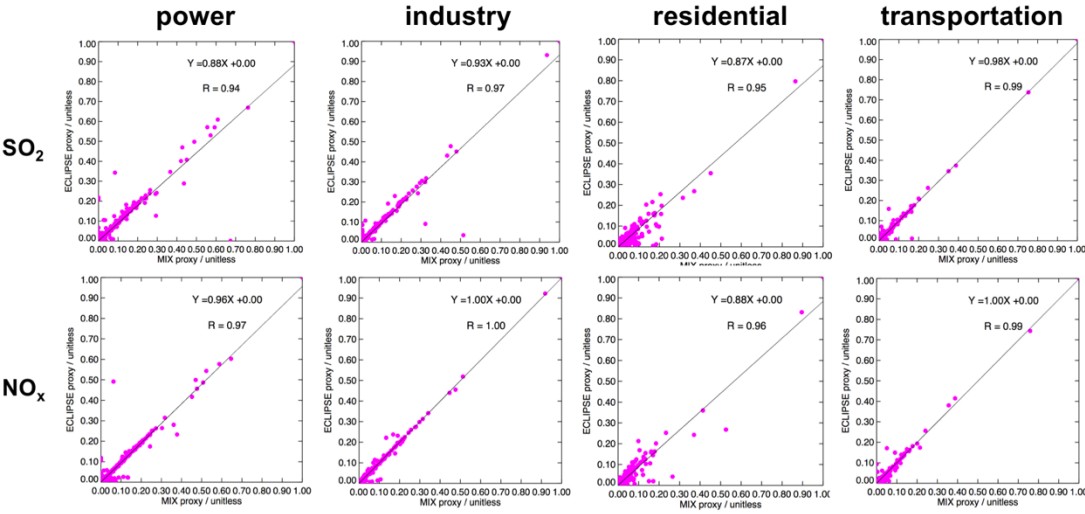

**Figure 4. Emission distribution ratios within provinces in China in 2010, 0.5° × 0.5° resolution.**

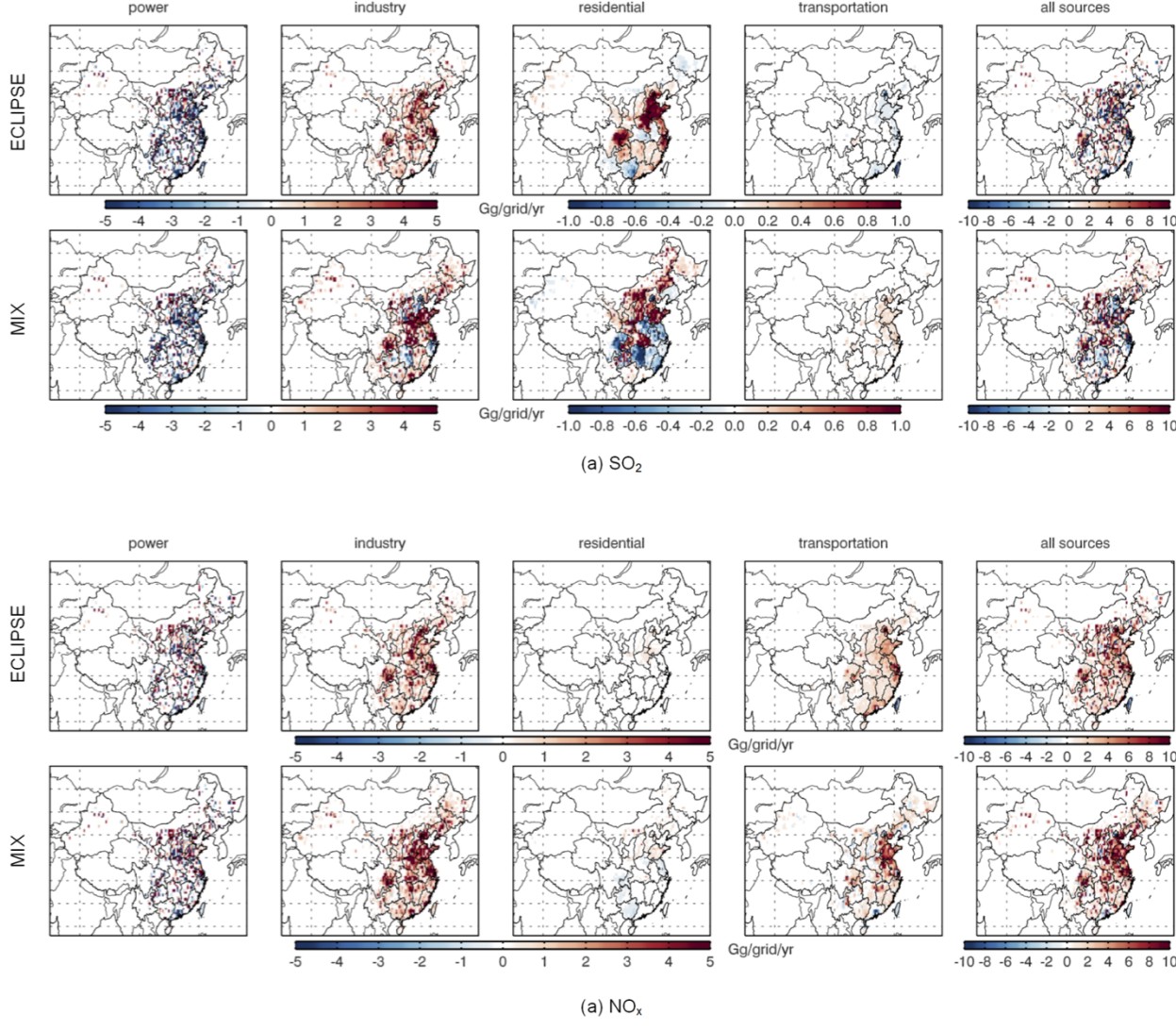

**Figure 5.** SO$_2$ (a) and NO$_x$ (b) emission changes from 2005 to 2010 by sectors, calculated as: $E_{2010} - E_{2005}$, at 0.5° × 0.5° grids. Note different color scales are used by sectors.

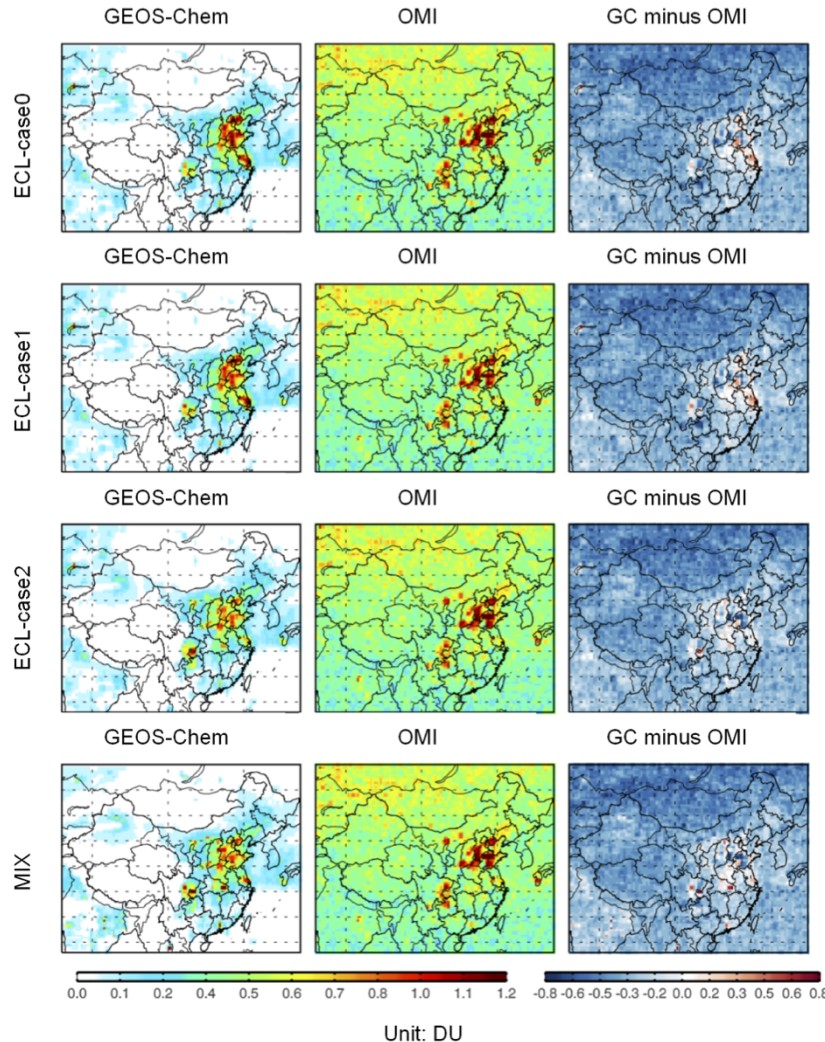

**Figure 6. SO₂ columns simulated by GEOS-Chem in sensitivity cases, compared to OMI observations, summer average (June-July-August) in 2010. Unit: 1DU=2.69×10¹⁶molec/cm².**

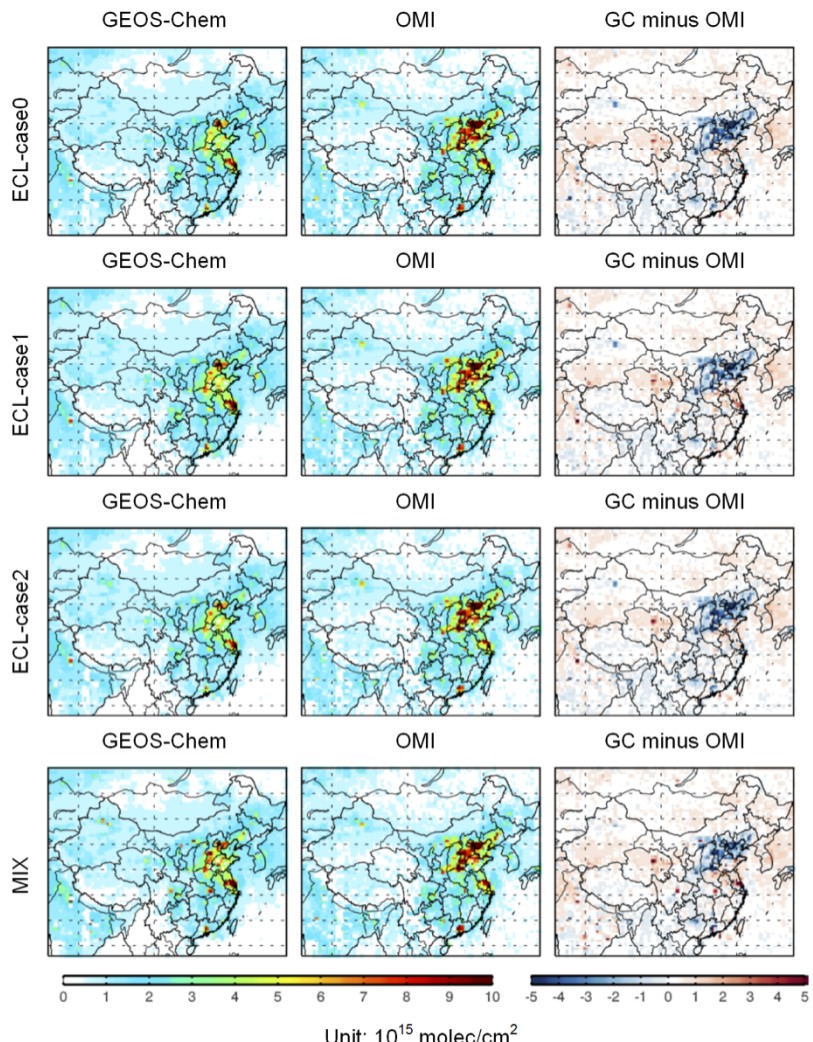

**Figure 7. NO$_2$ tropospheric columns simulated by GEOS-Chem in sensitivity cases. For each case, the specific NO$_2$ vertical profiles were applied in inversion, summer average (June-July-August) in 2010.**

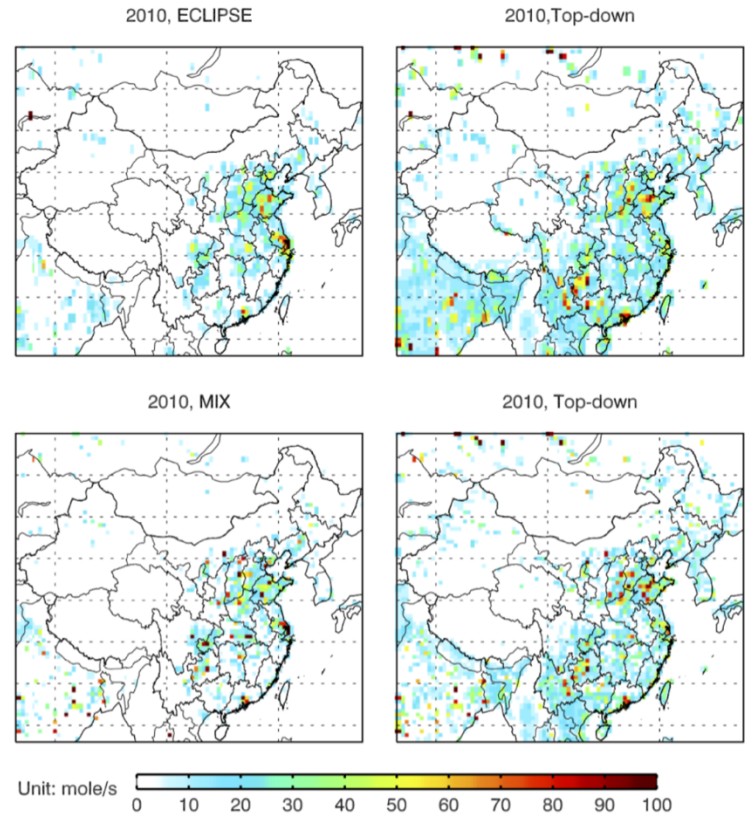

(a) SO₂ emissions in bottom-up and top-down inventories

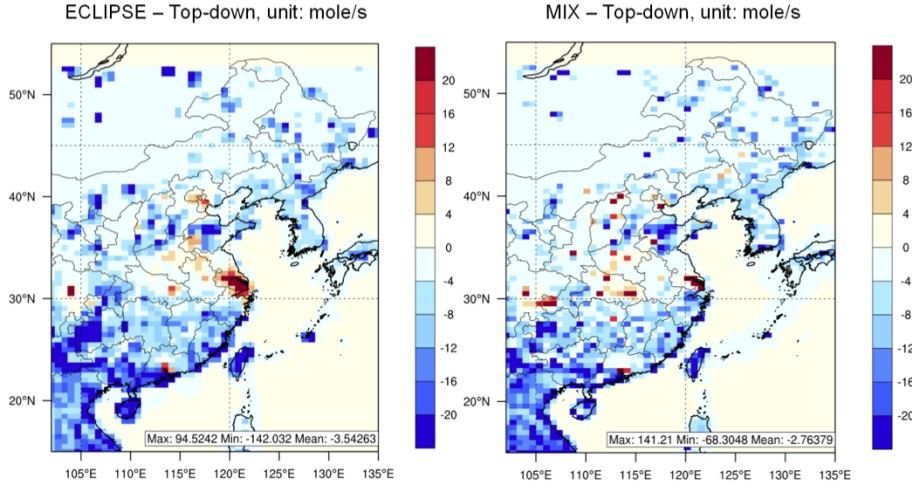

(b) Bottom-up minus Top-down SO₂ emissions, zooming in Eastern China

**Figure 8. Comparisons between ECLIPSE, MIX and top-down emissions of SO₂, summer average in 2010.**

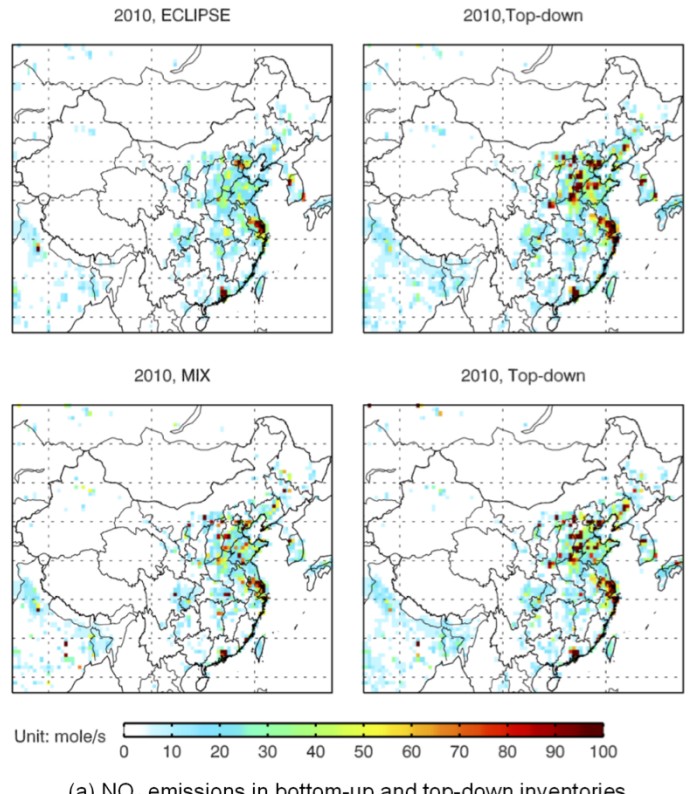

(a) NO$_x$ emissions in bottom-up and top-down inventories

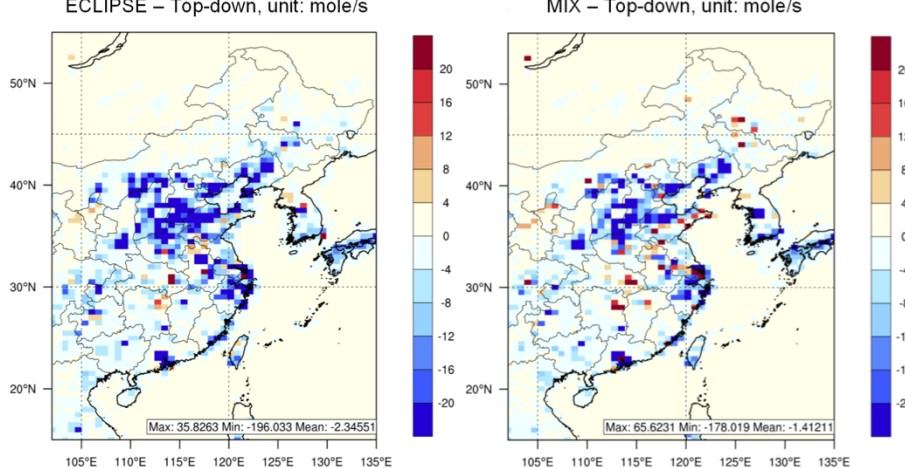

(b) Bottom-up minus Top-down emissions, zooming in Eastern China

**Figure 9. Comparisons between ECLIPSE, MIX and top-down emissions of NO$_x$, summer average in 2010.**

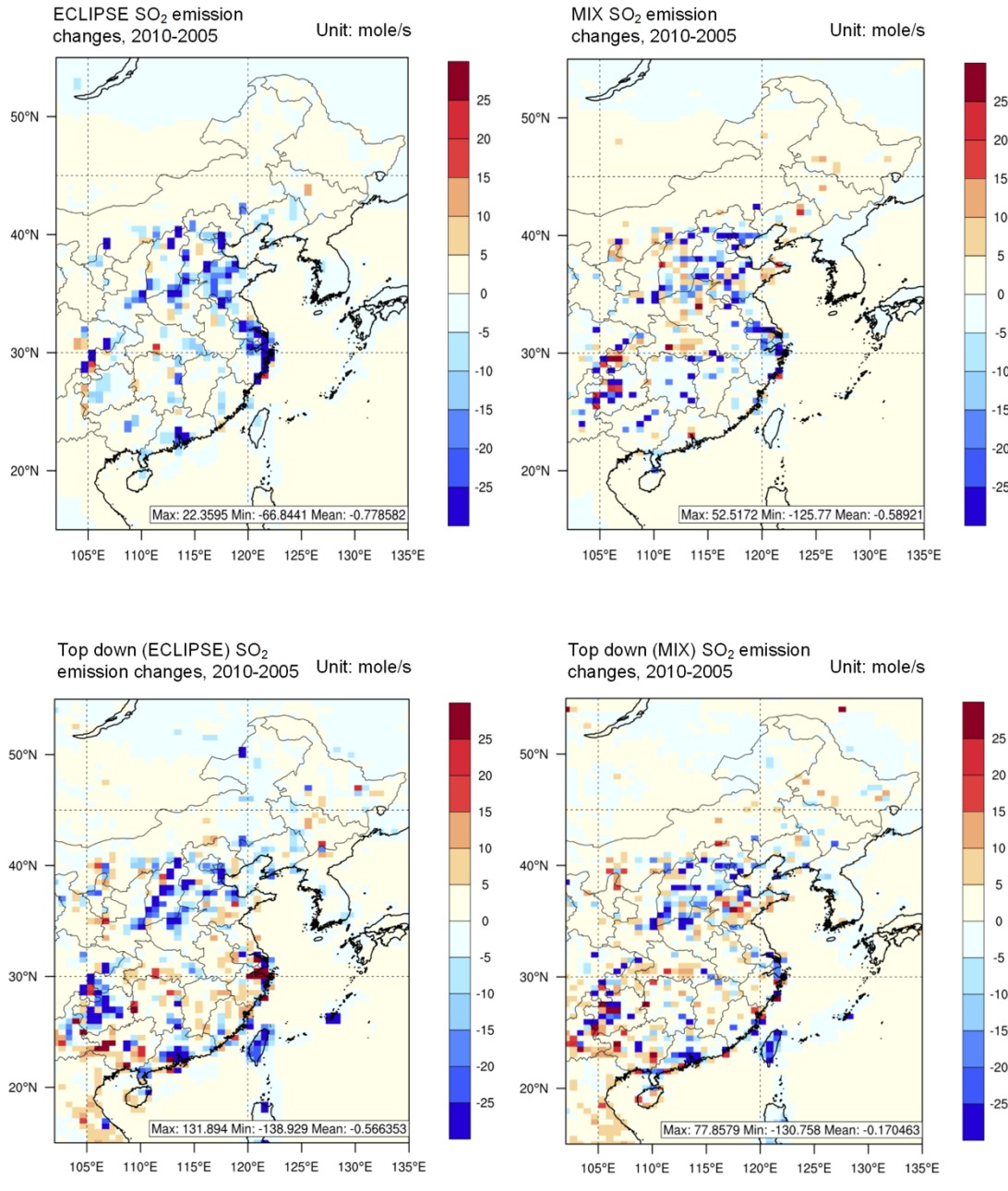

**Figure 10. SO$_2$ emission changes from 2005 to 2010, estimated by ECLIPSE, MIX, and derived top-down inventory for each case, summer average (JJA).**

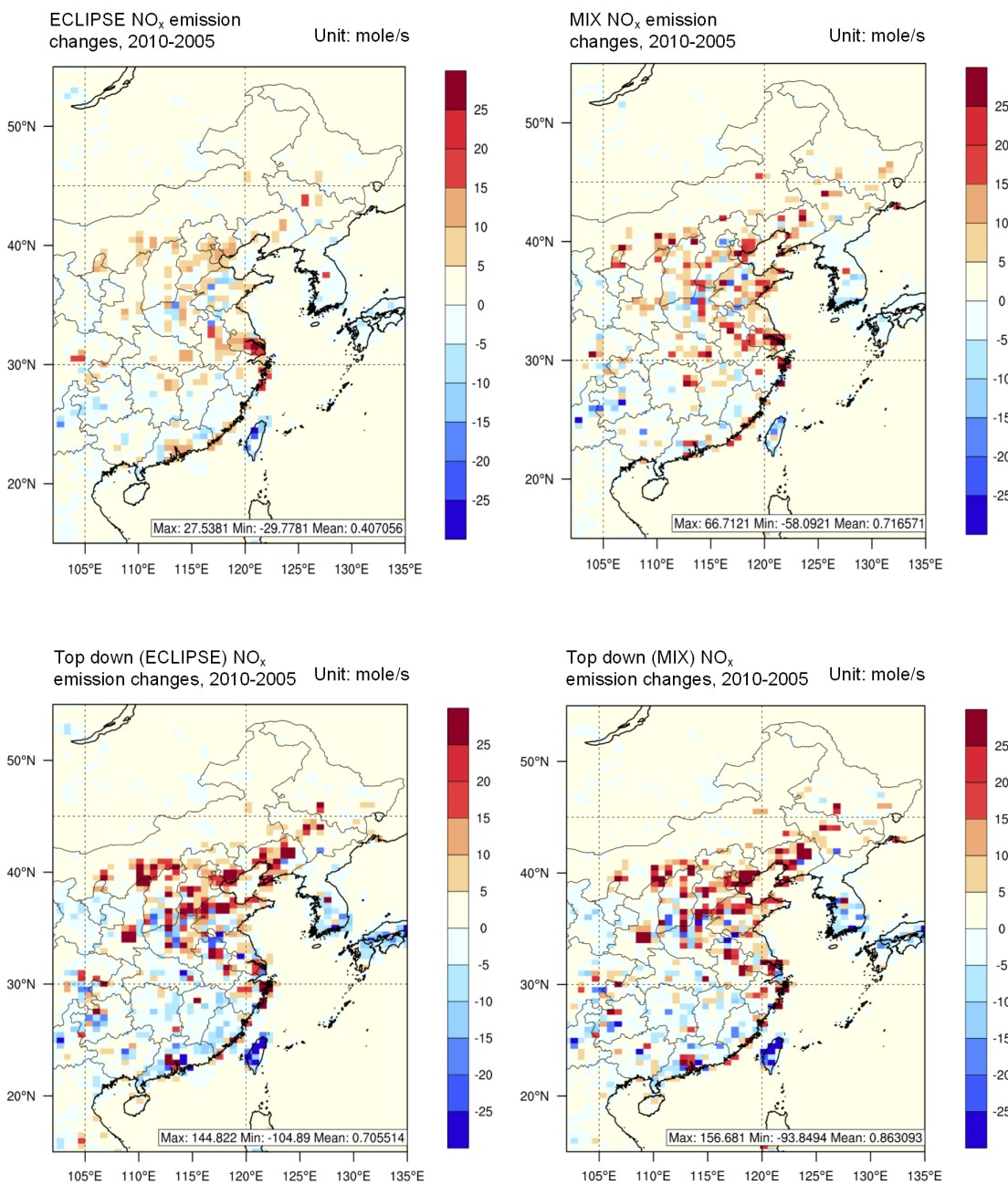

**Figure 11. NOₓ emission changes from 2005 to 2010, estimated by ECLIPSE, MIX, and derived top-down inventories for each case, summer average (JJA).**

