# Peer review of "Comparison and evaluation of anthropogenic emissions of $SO_2$ and $NO_x$ over China"

_Atmospheric Chemistry and Physics, 2017_

## Referee Comment (RC2) · Anonymous Referee #1 · 25 Aug 2017

This manuscript describes a comparison of two bottom-up SO2 and NOx inventories over China. The authors describe some of the input data, and explore the reasons for discrepancies between the two inventories. Satellite observations of NO2 are used with the GEOS-Chem model to produce top-down NOx emissions that are also used to evaluate the bottom-up inventories. The authors find that while differences in total emissions of SO2 and NOx from the bottom-up inventories are small, discrepancies at the sector level and provincial level are large. Compared to the top-down emissions, both bottom-up inventories are found to have negative biases, although uncertainty in the top-down approach cannot be ignored.

General comment:

This study is written clearly for the most part, and brings attention to specific de-

tails/uncertainties about bottom-up inventories that should be considered when used in chemical transport model simulations. In general, the methods are technically sound and the conclusions are supported by the results. However, as a reader I was left with a larger question: What is the take-home message of this article? What is its importance to the atmospheric chemistry community? The authors do a sound job of pointing out differences between two (seemingly arbitrary) bottom-up inventories, but besides the obvious conclusion that some inventories will be different than others as a result of different methods/datasets, I'm not sure of the relevance here. The manuscript is quite technical, and in my opinion, misses the mark in terms of scientific significance. I encourage the authors to consider how their results and conclusions have larger impact. As written, it's not clear what substantial new concepts or methods have been advanced.

Specific comments:

Abstract:

1) As written, the abstract seems to focus quite a bit on the methods, and very little on the results and relevance. I encourage the authors to consider editing their abstract to include the important results and conclusions.

Methodology:

1) What is the reason for focusing on ECLIPSE and MIX? It comes across as an arbitrary choice of inventories. Are they the most recently developed? Are they the most popular in chemical transport models? Do they provide the most methodological details? Why should the readers be interested in these two inventories specifically?

2) The spatial proxies are mentioned very generally many times throughout the manuscript, but almost no detail is given the methods about the actual data used in each case. On Page 14 the authors state, "Proxies used.. are summarized in Section 2". But unless I missed it entirely, they are not summarized beyond some very general

language. Further broad strokes are given on Page 14 ("for industry and residential sector, emissions are distributed mainly on population data. Road networks and population are used as proxy for transportation emissions"), but I think at the very least, these details belong in the methods earlier on. I was frustrated by the number of times spatial proxy data are referred to with general language ("mainly"; "including"), but did not come away with a comprehensive understanding. Can the authors include a table that summarizes the source of all the spatial proxy data that is actually used in each inventory/sector? Or perhaps include maps of the different spatial data used in the Supplementary Information? A lot of attention is given to the spatial patterns, for them to be of such little importance in the methods.

3) The authors point out that OMI SO2 observations have large uncertainties. Would the observations be at all valuable in a qualitative comparison of spatial emission patterns?

Results:

1) P 14 Line 2 mentions how the industrial and residential sectors show "clear administrative boundaries". But for someone who is not familiar with the administrative boundaries, this isn't obvious (Provincial? county?). Would it be useful to include some of the boundaries they are referring to?

2) P 14 Line 17 mentions how "other" proxies are population-based. Which proxies exactly are the authors referring to?

3) P 14 Line 20 mentions the excellent correlations observed for all sectors, but then misses the most interesting question. What are the exact sources of the occasions when they are different? For example, residential NOx has a slope of 0.88, whereas the slopes for the other sectors are all very close to 1. What data has been used differently that causes this difference in the residential sector between the two?

4) P 15 Line 15 "In light of the bottom-up comparisons". Here, can the authors be

specific about what issues they are referring to? Exactly what hypotheses are the sensitivity tests set up to test? This would help understand the importance and purpose of the sensitivity simulations.

5) P 18 Line 26: This is the first indication in the entire article that IGDP is used as a spatial proxy. This is a good example of why the discussion about spatial proxies became frustrating to me. Again, I encourage the authors to lay out or list the spatial proxies comprehensively in the methods. Perhaps these details are obvious to some, but they aren't obvious to me.

Figure 1: Might I suggest the authors include the totals for SO2 and NOx from each inventory in the figure (just as a number, somewhere in the panel)?

---

## Author Comment (AC1) · 22 Jan 2018

**Response to Referee #2:**

*General comments*

*The work of Li et al. deals with the comparison of the ECLIPSE and MIX emission inventories over China focusing on SO2 and NOx sector- and region- specific emissions. Bottom up emissions are then compared with top down estimates from OMI. The paper is overall well written and I recommend it for publication after developing the following points:*

*The authors should clarify the aim of their work, since comparing two emission inventories (even at sector level) and top down vs. bottom up estimates comparison are not new topics in literature. It is not completely clear the novelty of this work compared to literature studies dealing with top down and bottom up estimates such as Wang et al. 2011 and other works. The authors state that "To our knowledge, it's the first emission inventory assessment work where parameter-level comparison and remote sensing evaluations are combined", however, there are several literature works com- paring top down and bottom up estimates, even over China (e.g. Wang et al., 2011; R. J. van der A, 2017 etc.). Therefore, the authors should clarify the relevance of their study compared to former works.*

**Response:** We thank reviewer #2 for the careful reading and constructive comments, which are crucial to improve the manuscript.

As stated by the referee, several work conducted the emissions comparisons and top-down validations (such as Wang et al., 2011 from the same group). Extensive comparisons among emission inventories were important to illustrate the effect of emission inventory on the model simulation results and atmospheric component analyses (e.g., Ding et al., 2017; Saikawa et al., 2017). Although they provide important indications on the extent of discrepancies, there are still gaps for applying the comparison results to improve the inventory accuracy:

(1) Comparisons have been conducted for the total anthropogenic sources, instead of by sectors/subsectors/sources. Inconsistency of source categories included in inventory models were not overviewed or analyzed;

(2) Few studies go into the comparisons on specific parameter level because the technology-based framework for each inventory was not publicly available;

(3) Top-down and bottom-up comparisons have not been comprehensively combined to infer the potential uncertain parameters for all key sectors.

Through international collaborations between IIASA and Tsinghua University, we compared the ECLIPSE and MIX emissions over China parameter by parameters at a detailed activity-source level. What we focused on in this manuscript is the bottom-up comparison detailed to specific parameter contributing to the differences between the two widely used gridded emission inventories (ECLIPSE and MIX), combined with top-down validations from the satellite observations.

Another motivation of this work is to discuss the "fitness" of current developed inventories (specifically ECLIPSE and MIX) and modeling work done with them for policy relevant discussion. The inventories and the relevant modeling work is playing increasingly important

role for policy discussion in Europe and most recently more and more in Asia at different scales. However, there is no systematic and officially approved methods and inventories but a variety of scientific products. While a lot of effort has been made to validate emission estimates with measurements, higher source and spatial resolution of inventories and projections will serve also discussion about the how to shape future policies to reduce impact of air pollution.

We clarified the aim of this work in the revised manuscript.

The responses to the specific comments are provided below.

*Specific comments*

*- In the introduction the authors list several emission inventories covering China, how- ever, several other emission inventories have been developed for that region (e.g. Liu et al., 2015; the EDGAR database, etc.). The authors should explain why they provide only that list of references.*

**Response:** In the revised manuscript, we complemented the reference list by including all emission inventories that developed gridded emissions of $SO_2$ or $NO_x$ from anthropogenic sources over China as follows:

To support chemical transport modeling and provide scientific basis for policy-making, several emission inventories covering China have been developed (Streets et al., 2003; Ohara et al., 2007; Zhang et al.,2007, 2009; Lu et al., 2010, 2011; Kurokawa et al., 2013; Klimont et al., 2009, 2013; Wang et al., 2014; Li et al., 2017; EDGAR v4.2 (available at http://edgar.jrc.ec.europa.eu)).

*- 2.1 The ECLIPSE and MIX emission inventory: This paragraph describes the two inventories later compared in the paper. To facilitate such comparison, it would be good to have a summary table listing for the two inventories the data sources for each sector (activity data and emission factors), the temporal and spatial resolution, the reference years, compounds, etc. The authors should highlight the independence of the two inventories in terms of statistics, EFs, proxies, etc. before doing the comparison.*

**Response:** Thanks for the suggestions. We add a table listing the data sources of activity data, emission factors, and key features of ECLIPSE and MIX, to facilitate the comparison. We insert a paragraph before the detailed comparison by sectors in Sect. 3.1 as below:

"As shown in Table 1, the activity rates were assigned independently by two inventories. As a global emission inventory, ECLIPSE mainly relies on international statistics of IEA. Differently, MIX obtains the official statistics of energy consumption and industrial output from NBS (National Bureau of Statistics) or MEP (Ministry of Environmental Protection) of China. We can expect high independency for the determination of emission factors between ECLIPSE (GAINS model) and MIX (MEIC model). As two independently developed inventory model, the source classification, technology penetration, removal efficiencies in GAINS and MEIC are expected to be different although they both refer to up-to-date measurements and peer-reviewed

data. Different methods were developed in two inventory models for specific sectors, including power plants, transportation, and agriculture. For power plants, the spatial proxies were essentially consistent between ECLIPSE and MIX. For other sectors, emissions were gridded independently by two emission inventories (see Table S1)."

**Table 1. Key features of ECLIPSE v5a and MIX emission inventories.**

| Item | ECLIPSE v5a | MIX |
|---|---|---|
| Year | 1990-2010 at a 5-year interval | 2005[a], 2008, 2010 |
| Domain | Global | Asia |
| Spatial resolution | 0.5°×0.5° | 0.25°×0.25° |
| Temporal resolution | Monthly | Monthly |
| Activities included for each sector | | |
| Energy / Power | Power plants (including CHP), energy production/conversion (including district heating plants), fossil fuel distribution | Power plants (including CHP) |
| Industry | Industrial combustion and processes | Industrial combustion (including industrial heating plants) and industrial processes |
| Residential | Residential combustion sources | Residential combustion sources (including residential heating plants) |
| Transportation | On-road and off-road transport sources [b] | On-road and off-road transport sources [b] |
| Agriculture | Livestock and fertilization | Livestock and fertilization |
| Data sources of activity rates | | |
| Power | International Energy Agency (IEA) | CPED (Liu et al., 2015) |
| Industry | International Energy Agency (IEA) | Provincial industrial economy statistics (NBS) |
| Residential | International Energy Agency (IEA) | Provincial energy statistics (NBS) |
| Transportation | International Energy Agency (IEA) | Provincial energy statistics (NBS); Zheng et al. (2014) |
| Agriculture | UN Food and Agriculture Organization [c] | Provincial statistics (NBS, Huang et al., 2012) |
| Emission factors and technology | GAINS model (Klimont et al., 2017) | MEIC model [d], Process-based model for $NH_3$ (Huang et al., 2012) |
| Data Access | http://www.iiasa.ac.at/web/home/research/researchPrograms/air/ECLIPSEv5a.html | http://www.meicmodel.org/dataset- mix |

[a] developed following the same methodology

[a] International air and international shipping are not included.

[c] FAO, http://www.fao.org/faostat/en/#home.

[d] Zhang et al., 2009; Lei et al., 2011; Zheng et al., 2014; Liu et al., 2015

*- 2.3 Top-down emission inventory: The authors should explain why the methodology presented is applied only to NOx and not to SO2 columns. It would be interesting to see the same procedure applied also to SO2 since the paper focuses on both compounds.*

**Response:** Thanks for the suggestions. We add more illustrations of top-down validations for $SO_2$ emissions. Firstly, we compared the $SO_2$ concentrations between modeled results and OMI $SO_2$ columns. Although OMI data tend to overestimate the concentrations due to the overlap in signals of $SO_2$ and $O_3$ during retrieval, good correlations are found between models and satellite (R=0.633-

0.667, Slope=0.842-0.863, general consistent by sensitivity cases), confirming the high accuracies of the priori $SO_2$ spatial emission patterns.

Secondly, following the method of $NO_2$ top-down inversion, we developed the top-down emissions for $SO_2$. The spatial distributions, emission amount and trend of the priori emission inventory were further evaluated. Given the large uncertainties involved in the OMI $SO_2$ columns, the evaluated results were interpreted with caution. Compared to the top-down emission inventory, both ECLIPSE and MIX show relatively good correlations (R=0.722-0.896, Slope = 0.539-0.923). The national decrease trend from 2005 to 2010 were captured by both bottom-up and top-down inventories.

Figures, tables and discussions of top-down evaluations for $SO_2$ were added in the revised manuscript.

*- Page 6, line 4: please clarify how the sectors "power", "industry", "residential" and "transportation" are defined for each inventory. As described at lines 7-10, sectors are different for the two inventories. Please clarify how emissions from heating plants are re-distributed (line 9) in MIX to match the ECLIPSE sectors.*

**Response:** We clarified the definition of each sector in Table 1 (see above). We aggregated the heating emissions from the "industry" and "residential" sectors in MIX to the "power" sector, to match the defined ECLIPSE sectors. We clarified the procedure in the revised manuscript.

*-page 10, line 8: "emission factors on mass base are converted to energy base with heating value of 43.1 MJ/kg". Did the authors use the same heating value both for gasoline and diesel?*

**Response:** Yes. These values are extracted from the GAINS model, comparable to the reported heating values of 44-46 MJ/kg for gasoline and 45 MJ/kg for diesel fuel by the World Nuclear Association (http://www.world-nuclear.org/information-library/facts-and-figures/heat-values-of-various-fuels.aspx). We revised the note for clarification.

*-page 10, line 10: although only 3% difference is found in total gasoline consumption, big differences in gasoline use by vehicle are observed for the two inventories.*

**Response:** We add more illustrations by vehicle types as follows:

"The consistency in the total gasoline consumption between ECLIPSE and MIX is attributed to the consistency in statistics. As shown in Table 3, the gasoline consumptions by vehicle types show large differences between ECLIPSE and MIX, indicating different vehicle fleet assumption in two inventory models. Detailed data is not known and each of the inventories (or research groups developing them) relied on own assumptions about fuel consumption per vehicle, mileage travelled, and combined those with the available data on the number of vehicles, their sales and retirement rate. Owing to the above reasons, the results can differ significantly. Light duty vehicles are the largest gasoline consumer (> 77%) in both inventories, with 18% higher gasoline consumption estimated in ECLIPSE than those of MIX in 2010.

Accordingly, ECLIPSE estimates less gasoline consumed in high duty vehicles (74%) and motorcycles (32%) than MIX. These differences reduced from 2005 to 2010."

*-page 10, line 11: huge differences are observed not only for light duty vehicles but also for HDV-G and MC.*

**Response:** We focused on the comparison for light duty vehicles because they dominate the total emissions of gasoline-fueled vehicles, with emission contributions of 63% - 91% (estimated by two inventories) in 2010. For HDV-G and MC, we extended the discussion in the revised manuscript as below:

"Emission estimates of HDV-G (high duty vehicles) and MC (motorcycles) also show large differences between two inventories. For HDV-G, ECLIPSE estimates lower emissions than MIX (66% in 2010), as a result of less fuel consumption while higher emission factors in ECLIPSE. For MC, emissions of ECLIPSE are 64% lower than MIX, contributed by both fuel consumption and emission factors, as shown in Table 3."

*-3.1.3 Gridded emissions: Figure 3b shows the difference of the ECLIPSE-MIX gridded emissions. Did the authors compare the proxy data used by the two inventories to grid the emissions? A mismatch in the location of large point sources as well as the application of weighting factors to redistribute the emissions could strongly affect this type of calculation. Please develop this topic.*

**Response:** The differences of gridded emissions illustrated in Fig. 3 are attributed to the discrepancies in emission estimates nationwide and by provinces (Sect. 3.1.1, Sect. 3.1.2), and also method and data in emission spatial allocations (see Sect. 2). We add a table (Table S1) summarizing the spatial proxy data used in both inventories in the revised manuscript. For power plants which were treated as point sources, emissions are gridded based on the locations verified by Google Earth (Liu et al., 2015), consistent between ECLIPSE and MIX. For other sectors, ECLIPSE gridded the provincial emissions according to the source-specific layers, and MEIC used two-step allocation method (province to county, county to grid). The data sources of spatial proxies also differ between two inventories. In general, as illustrated in Figure 4, the spatial distributions of emissions within provinces show quite good consistency, even by sectors. We revised Sect. 3.1.3 to clarify it.

*- page 15, lines 7-9: "The different trends of transportation emissions are attributed to the different assumptions on legislation effect on pollution control in two inventory systems". The authors should demonstrate the aforementioned statement.*

**Response:** We complemented the discussion as follows:

"For Beijing, the differences of emissions trend in the transportation sector are mainly caused by diesel vehicles. In ECLIPSE, 47% increases are estimated for diesel fueled vehicles, compared to 28% emission decreases in MIX. Fuel consumptions show large discrepancies in

trend from 2005 to 2010, where +54% compared to -20% for high duty vehicles, and +45% compared to +3% for light duty vehicles, as estimated in ECLIPSE and MIX respectively. The emission factors of light duty vehicles increase by 5% in ECLIPSE, while decrease by 34% in MIX, attributed to the different assumptions on emission control effects. As a pioneer in pollution control of China, Beijing carried out Euro III standard in 2005 and Euro IV standard in 2008 for light duty vehicles. For Beijing, the Euro IV penetrations in 2010 are assumed around 12% in ECLIPSE, while more than 60% in MIX, which might be too optimistic and should be verified with local surveys.

For the PRD region, gasoline and high duty diesel vehicles contribute to the differences of emission trend. 22% emission growth for LDB-G (light duty gasoline buses) is estimated in ECLIPSE, compared to 12% emission reduction in MIX. For high duty diesel vehicles, trend of fuel consumption (+55% in ECLIPSE, compared to -11% in MIX) and technology distribution (21% of Euro III in 2010 for ECLIPSE, compared to >50% in MIX) are the main contributors to the different emissions trend. In summary, survey data are urgently needed to validate the fuel consumptions, effect of legislation effect and trend for diesel vehicles in pioneering regions such as Beijing and PRD."

*- page 18, line 17: "It can be concluded that ECLIPSE and MIX are consistent with the top-down estimates over China." The authors should discuss why it is useful to compare bottom up and top down estimates. In their work they discuss the differences (sometimes not negligible) between two bottom up inventories over China and then through the comparison with top down estimates they find that the two inventories are consistent with these independent estimates. How is that possible? How can top down estimates help in constraining the bottom up emission inventories? How can this work reduce the uncertainty of emission inventories? Can the authors explain if the uncertainty of bottom up and top down estimates are larger, smaller or within the range of model uncertainties?*

**Response:** Top-down emission estimates can provide independent third-party constraints on the bottom-up emissions on the emissions amount, spatial distribution and trend (Martin et al., 2003; Lamsal et al., 2011; Lee et al., 2011; Liu et al., 2016; Cooper et al., 2017). In this work, we compared the detailed parameters for bottom-up emission inventory development in the previous sections, and found general consistent emission estimates for the whole China (differ within 16%) and gridded emissions (slope ≥ 0.8, R ≥ 0.9), while large variations for provincial emissions and specific sectors. These detailed comparisons are important for guiding the emission inventory community to put more efforts on parameters that are quite uncertain in current inventory system, including the real-world running status of pollutant abatement facilities, statistics of diesel consumption, vehicle fleet, and emission factors of industrial boilers.

Gridded emissions are the direct inputs to atmospheric models. Through comparing the gridded emissions from bottom-up and top-down emissions derived from satellite observations, the model-ready emissions input can be overall constrained, given the comparable uncertainties in two inventories (as illustrated in Sect. 3.3.2). Compared to the top-down emissions, both ECLIPSE and MIX show high correlations (R ≥ 0.87), supporting the conclusion that both

inventories are generally consistent with the top-down estimates in spatial emission patterns. We also found moderate negative biases (-21% - 39%) for the bottom-up inventories, indicating that ECLIPSE and MIX may underestimate $NO_x$ emissions in 2010, which indicates the direction of verifying the uncertain parameters. Furthermore, the emission trends were validated based on the top-down retrievals. The general consistent emission trend proves the "fitness" of our inventories for policy relevant discussion and projections.

*- page 19, lines 1-2: "Through sensitivity test analyses, treating sources as point sources can significantly reduce the uncertainties in emission gridding process". The authors should better explain how it is possible to reduce the uncertainties in emission gridding process through sensitivity tests. Sensitivity tests can help understanding the uncertainties due to the gridding procedure using e.g. different proxy data, but not necessarily to reduce the corresponding uncertainty.*

**Response:** We refer to the conclusions of Geng et al. (2017) here. In the work of Geng et al. (2017), a set of sensitivity test was conducted to evaluate the impact of spatial proxies on model performance. It's proved that determining the exact locations of large emission sources will significantly strengthen the correlation with modeled and satellite retrieved $NO_2$ columns (Geng et al., 2017). To avoid misunderstanding, we revised the sentence to "Through sensitivity test analyses, it's concluded that treating sources as point sources can significantly reduce the uncertainties in emission gridding process (Geng et al., 2017)".

*- It would be interesting to see Figure S1 also in absolute terms. The authors should also better explain the different sectorial share for the various provinces. Why Tibet has only SO2 emissions from the transportation sector in the MIX inventory, while they are negligible for ECLIPSE? Large sector specific differences are also observed for NOx. Please discuss in a more comprehensive way the differences in sector specific emissions at province level.*

**Response:** The sectorial distributions of provincial emissions are overall consistent (within 30% difference on sector level) given the differences in source classification between two inventory systems. For Tibet, the emissions can be neglected (e.g., for $NO_x$, around 30 Gg/year, 0.1% of the national total) and unreliable for both emission inventories because the real-world energy consumption statistics are quite uncertain. We add the emissions by provinces in Figure S1 to give a more comprehensive reference to readers. The absolute values of sectorial emission share for each province are labeled in Figure S1.

*Technical corrections*

*- Figure 5a shows empty maps for the SO2 trend from transportation sector of both inventories. Please check them.*

**Response:** The $SO_2$ emissions from the transportation sector are ignorable compared to other sectors under the same color scale. We revised Figure 5a with different color scale for the

transportation sector.

-*Figure S3: Please change the Figure caption with "NOx emission changes. . ." instead of "Emission changes. . .".*

**Response:** Revised as suggested.

[revised manuscript text omitted]

---

## Author Comment (AC2)

**Response to Referee #1:**

*This manuscript describes a comparison of two bottom-up SO2 and NOx inventories over China. The authors describe some of the input data, and explore the reasons for discrepancies between the two inventories. Satellite observations of NO2 are used with the GEOS-Chem model to produce top-down NOx emissions that are also used to evaluate the bottom-up inventories. The authors find that while differences in total emissions of SO2 and NOx from the bottom-up inventories are small, discrepancies at the sector level and provincial level are large. Compared to the top-down emissions, both bottom-up inventories are found to have negative biases, although uncertainty in the top-down approach cannot be ignored.*

**Response:** We thank the constructive comments from reviewer #1 to improve our manuscript. Detailed responses to each comments are provided below.

*General comment:*

*This study is written clearly for the most part, and brings attention to specific details/uncertainties about bottom-up inventories that should be considered when used in chemical transport model simulations. In general, the methods are technically sound and the conclusions are supported by the results. However, as a reader I was left with a larger question: What is the take-home message of this article? What is its importance to the atmospheric chemistry community? The authors do a sound job of pointing out differences between two (seemingly arbitrary) bottom-up inventories, but besides the obvious conclusion that some inventories will be different than others as a result of different methods/datasets, I'm not sure of the relevance here. The manuscript is quite technical, and in my opinion, misses the mark in terms of scientific significance. I encourage the authors to consider how their results and conclusions have larger impact. As written, it's not clear what substantial new concepts or methods have been advanced.*

**Response:** Thanks for the reviewer's suggestions.

As for the aim of this work, as illustrated in the text, we want to better understand the reasons of differences between inventories for the currently widely used emission inventories (especially the two were picked because we have good access of the full inventory model, data and results), evaluate their effects on model simulation and their accuracies from the remote sensing perspective. The reasons of selecting ECLIPSE and MIX are detailed illustrated in the response to specific comments below. Although inventory comparisons have been conducted by several studies, there are still gasps to apply these results to improve the inventory accuracy. The comparisons cannot point to the systematic uncertainty of source, parameters, assumptions in inventories because of the lacking of the detailed technology-based inventory model. Initiated by this, we performed the comparisons and analyses of the two inventory model on parameter level to give more explicit indications for inventory developers. Our study should be important for inventory developers and modelers for understanding the potential uncertainties in the gridded emission inventory over China. For modelers, the comparisons and validations are important to understand the effect of emissions on model performance.

Another motivation of this work is to discuss the "fitness" of current developed inventories

(specifically ECLIPSE and MIX) and modeling work done with them for policy relevant discussion. The inventories and the relevant modeling work is playing increasingly important role for policy discussion in Europe and most recently more and more in Asia at different scales. However, there is no systematic and officially approved methods and inventories but a variety of scientific products. While a lot of effort has been made to validate emission estimates with measurements, higher source and spatial resolution of inventories and projections will serve also discussion about the how to shape future policies to reduce the impact of air pollution. In fact, this discussion already takes place and science contributes to it. This work shows that our best inventories appear to be fit for evaluation of the policies at an aggregated or national level, more work is needed in specific areas in order to improve accuracy and robustness of outcomes at the finer spatial and also technology level.

The main take-home messages are illustrated in the "Concluding Remarks" section. Generally, this work shows that our best inventories are consistent on emission estimates at national level. The spatial distribution, emission amount and trend of current inventories are well captured by the satellite remote sensing. It appears that current inventories and relevant modeling work are fit for policy analyses and future projections at an aggregated level. More work is needed to improve the accuracy through verifying important while currently uncertain parameters on finer spatial units, including the real-world running efficiency and application rate of pollutant control facilities of FGD and LNB, diesel consumptions, vehicle fleet and emission factors for various vehicle types.

Specifically, the take-home messages are:

a)  ECLIPSE and MIX estimates of $SO_2$ and $NO_x$ are quite close in 2010, with 1% and 16% difference in China. The trends from 2005 to 2010 estimated by two inventories are both generally consistent with the OMI observations.

b)  The emission differences on provincial and sector level are still high. On a sector level, 40% difference is found for power plants (higher in ECLIPSE), 24% for the industry sector (lower in ECLIPSE) for $SO_2$, and 15%~21% in power and transportation for $NO_x$ (lower in ECLIPSE). Source classification, energy statistics, emission factors and assumption of technology penetrations are contributing parameters. Specifically, the FGD penetration rate and removal efficiencies, the LNB application rates and abatement efficiency in power plants, vehicle fleet and emission factors for various vehicle types in the transportation sector need further verification. Diesel consumptions are quite uncertain, which need more local surveys and validations for future inventory improvement.

c)  The model case done with MIX show the best performance. Increasing the ECLIPSE emission estimates to MIX reduces the biases from -12.2% to -6.19% for $NO_x$. Negative biases in bottom-up gridded emission inventories are found compared to the top-down inversion.

We addressed the aim and importance of our work in the introduction section. We clarified the take-home message points in the concluding section, and revised the abstract to highlight the main findings.

*Abstract:*

*1) As written, the abstract seems to focus quite a bit on the methods, and very little on the results and relevance. I encourage the authors to consider editing their abstract to include the important results and conclusions.*

**Response:** Thanks for the suggestions. We revised the abstract to include important results, relevance to the atmospheric community and main findings.

*Methodology:*

*1) What is the reason for focusing on ECLIPSE and MIX? It comes across as an arbitrary choice of inventories. Are they the most recently developed? Are they the most popular in chemical transport models? Do they provide the most methodological de-tails? Why should the readers be interested in these two inventories specifically?*

**Response:** We focused on the comparisons of ECLIPSE and MIX due to the following reasons:

a) Up to the time of manuscript preparation, ECLIPSE and MIX are the only publicly accessible gridded emissions dataset which include both $SO_2$ and $NO_x$ covering China for the period of 2005 and 2010;

b) Both inventories have been widely applied in atmospheric modeling and policy discussions (e.g., Stohl et al., 2015; Duan et al., 2016; Galmarini et al., 2017; Rao et al., 2017);

c) The technology-based framework and compiling parameters by source categories are obtained for ECLIPSE and MIX through international collaboration, which is not accessible for other inventories over China. The methods and data were extensively described by a series of paper (Liu et al., 2015; Zheng et al., 2014; Klimont et al., 2017; Li et al., 2017), supporting us for explicit comparisons and analyses;

d) ECLIPSE (GAINS model) can be representative of the state-of-science global emission inventory covering China, and MIX (MEIC model) as the regional inventory compiled with advanced methods and local data. The methods, parameters and assumptions of GAINS and MEIC are always referred by inventory developers (e.g., Lu et al., 2010; Fu et al., 2013; Kurokawa et al., 2013; Zhao et al., 2013). The comparisons and validations are important to improve the accuracy of gridded emissions and model performance over China.

We clarified the reasons of comparing these two emission inventories in the revised manuscript.

*2) The spatial proxies are mentioned very generally many times throughout the manuscript, but almost no detail is given the methods about the actual data used in each case. On Page 14 the authors state, "Proxies used .. are summarized in Section 2". But unless I missed it entirely, they are not summarized beyond some very general language. Further broad strokes are given on Page 14 ("for industry and residential sector, emissions are distributed mainly on population*

*data. Road networks and population are used as proxy for transportation emissions"), but I think at the very least, these details belong in the methods earlier on. I was frustrated by the number of times spatial proxy data are referred to with general language ("mainly"; "including"), but did not come away with a comprehensive understanding. Can the authors include a table that summarizes the source of all the spatial proxy data that is actually used in each inventory/sector? Or perhaps include maps of the different spatial data used in the Supplementary Information? A lot of attention is given to the spatial patterns, for them to be of such little importance in the methods.*

**Response:** Thanks for the comments on the spatial proxies. We add a table summarizing the spatial proxies by sub-sectors for ECLIPSE and MIX in the supplement to avoid repeated presentation with previous work (Klimont et al., 2017; Li et al., 2017), and enrich the elaborations in the main text for clarification:

"Emissions are distributed to grids at specific resolution ($0.5° \times 0.5°$ for ECLIPSE, longitude$\times$latitude) based on the percentages of spatial proxies located in grids by source category using GIS (Geographic Information System) techniques. Spatial proxies for both ECLIPSE and MIX are summarized in Table S1. For ECLIPSE, several layers were developed as spatial proxies in line with those used in the Representative Concentration Pathways (RCP) (Lamarque et al., 2010), i.e., locations of energy and manufacturing facilities, road networks, shipping routes, human and animal population density and agricultural land use. Spatial proxies were further developed within the Global Energy Assessment project (GEA project, Riahi et al., 2012), including improved population distribution, flaring in oil and gas production, smelters, and power plants for which provincial emission layers of MEIC (Multi-resolution Emission Inventory for China) were used.

For MIX, provincial emissions were firstly distributed to county, then further distributed to grids. The former process was based on statistics by county (i.e., GDP, IGDP, total population, urban population, rural population, agricultural activity, vehicle population at county level), and the latter was based on gridded maps as spatial proxies (i.e., population density map, road network). For power plants, locations were determined using Google Earth following the unit-based methodology."

Table S1. Spatial proxies used in the ECLIPSE and MIX emission inventories.

| Sub-sectors | Spatial proxies | Data source |
|---|---|---|
| **ECLIPSE** | | |
| Power plants | Plant locations | Google Earth (Liu et al., 2015) |
| Industrial combustion [a] | Total population, urban population, rural population | Lamarque et al., 2010; Riahi et al., 2012 |
| Industrial processes [a] | Urban population, industrial plants | Lamarque et al., 2010; Riahi et al., 2012 |
| Residential [a] | Total population, urban population, rural population | Lamarque et al., 2010; Riahi et al., 2012 |
| On-road transportation [a] | Population, Road networks | Lamarque et al., 2010; Riahi et al., 2012 |
| Off-road transportation [a] | Inland waterways, roads, railways, population, urban population | Lamarque et al., 2010; Riahi et al., 2012 |
| Waste [a] | Population, urban and rural population | Lamarque et al., 2010; Riahi et al., 2012 |

| Agriculture (fertilizer) | Cropland area | Potter et al., 2010 |
| Agriculture (livestock) | Livestock map | FAO, 2007 |

| **MIX[b]** | | |
| --- | --- | --- |
| Power | Plant locations | Google Earth (Liu et al., 2015) |
| Industrial heating | Industrial GDP[c], urban population[d] | NBS[e], LandScan[f], urban/rural extents[g] |
| Residential heating | Urban population | NBS, LandScan, urban/rural extents |
| industrial boiler | Industrial GDP[c], urban population[d] | NBS, LandScan, urban/rural extents |
| Residential combustion (fossil fuel) | Urban population | NBS, LandScan, urban/rural extents |
| Residential combustion (biofuel) | Rural population | NBS, LandScan, urban/rural extents |
| Iron and steel | Industrial GDP[c], urban population[d] | NBS, LandScan, urban/rural extents |
| Cement | Industrial GDP[c], urban population[d] | NBS, LandScan, urban/rural extents |
| Other industrial process | Industrial GDP[c], urban population[d] | NBS, LandScan, urban/rural extents |
| On-road vehicles | Vehicle population[c], road network[d] | China Digital Road-network Map (Zheng et al., 2014) |
| motorcycles | Vehicle population[c], road network[d] | China Digital Road-network Map (Zheng et al., 2014) |
| Off-road (agriculture machinery) | Machine power[c], rural population[d] | NBS, LandScan, urban/rural extents |
| Off-road (construction) | Total GDP[c], urban population[d] | NBS, LandScan, urban/rural extents |
| off-road (others) | Total population | NBS, LandScan |
| Solvent use - industry | Industrial GDP[c], urban population[d] | NBS, LandScan, urban/rural extents |
| Solvent use - residential | Urban population | NBS, LandScan, urban/rural extents |
| Agriculture (fertilizer) | Fertilizer use[c], rural population[d] | NBS, LandScan, urban/rural extents |
| Agriculture (livestock) | Meat consumption[c], rural population[d] | NBS, LandScan, urban/rural extents |
| Waste | Total population | NBS, LandScan |

[a] Spatial proxies included were derived from the EDGAR emissions gridding manual,

http://publications.jrc.ec.europa.eu/repository/bitstream/JRC78261/edgarv4_manual_i_gridding_pubsy_final.pdf

[b] derived from Li et al. (2017)

[c] Proxies used to distribute provincial emissions to county

[d] Proxies used to distribute county-level emissions to grids

[e] National Bureau of Statistics, http://www.stats.gov.cn/tjsj/

[f] LandScan Global Population database

[g] Urban / rural extents developed by Schneider et al. (2009)

*3) The authors point out that OMI SO$_2$ observations have large uncertainties. Would the observations be at all valuable in a qualitative comparison of spatial emission patterns?*

**Response:** Thanks for the suggestions. We add more illustrations of top-down validations for SO$_2$ emissions. Firstly, we compared the SO$_2$ concentrations between modeled results and OMI SO$_2$ columns. Although OMI data tend to overestimate the concentrations due to the overlap in signals of SO$_2$ and O$_3$ during retrieval, good correlations are found between models and satellite

(R=0.633-0.667, Slope=0.842-0.863, general consistent among sensitivity cases), confirming the high accuracies of the priori $SO_2$ spatial emission patterns.

Secondly, following the method of $NO_2$ top-down inversion, we developed the top-down emissions for $SO_2$. The spatial distributions, emission amount and trend of the priori emission inventory were further evaluated. Given the large uncertainties involved in the OMI $SO_2$ columns, the evaluated results were interpreted with caution. Compared to the top-down emission inventory, both ECLIPSE and MIX show relatively good correlations (R=0.722-0.896, Slope = 0.539-0.923). The national decreasing emission trend from 2005 to 2010 were captured by both bottom-up and top-down inventories.

Figures, tables and discussions of the top-down evaluations for $SO_2$ are added in the revised manuscript.

*Results:*

*1) P 14 Line 2 mentions how the industrial and residential sectors show "clear administrative boundaries". But for someone who is not familiar with the administrative boundaries, this isn't obvious (Provincial? county?). Would it be useful to include some of the boundaries they are referring to?*

**Response:** We add the provincial boundaries for all gridded emission maps to make it more clear.

*2) P 14 Line 17 mentions how "other" proxies are population-based. Which proxies exactly are the authors referring to?*

**Response:** "Other proxies" refer to the total population map (for some industrial sources), urban population map (for industrial heating, residential coal burning, etc.) and rural population map (for residential biofuel burning). We clarified the sentence in the revised manuscript.

*3) P 14 Line 20 mentions the excellent correlations observed for all sectors, but then misses the most interesting question. What are the exact sources of the occasions when they are different? For example, residential NOx has a slope of 0.88, whereas the slopes for the other sectors are all very close to 1. What data has been used differently that causes this difference in the residential sector between the two?*

**Response:** The excellent correlations for all sectors (with slope ≥ 0.87, R ≥ 0.94) are attributed to the overall similarity in spatial proxies (as summarized in the table of spatial proxies). The differences for specific sectors (e.g., residential with slope of 0.87) are slightly higher than others, mainly due to the different population dataset used for emission allocation of relevant sources in ECLIPSE and MIX. We clarified it in the revised manuscript.

*4) P 15 Line 15 "In light of the bottom-up comparisons". Here, can the authors be specific about what issues they are referring to? Exactly what hypotheses are the sensitivity tests set up to test? This would help understand the importance and purpose of the sensitivity simulations.*

**Response:** The bottom-up comparisons work done earlier show that differences exit for both the national emission estimates and gridded emission spatial patterns. In Sect. 3.2.1, the main purpose is to evaluate the effect of gridded emissions on model accuracy, and find out the effect of the national emission estimates and spatial distributions on the model simulated results, using satellite observations as a criterion. Therefore, we further set up four sensitivity tests, ECL-case0~case2 and MIX, where ECL-case0 and MIX are two basic inventory scenario, ECL-case1 replaced the total emissions of ECL-case0 with MIX's estimates, and ECL-case2 changed the spatial distributions of ECL-case0 to MIX. We clarified this in the revised manuscript.

*5) P 18 Line 26: This is the first indication in the entire article that IGDP is used as a spatial proxy. This is a good example of why the discussion about spatial proxies became frustrating to me. Again, I encourage the authors to lay out or list the spatial proxies comprehensively in the methods. Perhaps these details are obvious to some, but they aren't obvious to me.*

**Response:** We summarized the spatial proxies in Table S1 in the revised manuscript.

*Figure 1: Might I suggest the authors include the totals for SO2 and NOx from each inventory in the figure (just as a number, somewhere in the panel)?*

**Response:** Revised as recommended.

[revised manuscript text omitted]